# Increasing Notch signaling antagonizes PRC2-mediated silencing to promote reprograming of germ cells into neurons

Stefanie Seelk[1†], Irene Adrian-Kalchhauser[2†‡], Balázs Hargitai[2],
Martina Hajduskova[1], Silvia Gutnik[2], Baris Tursun[1*], Rafal Ciosk[2*]

[1]Berlin Institute for Medical Systems Biology, Max-Delbrück-Center for Molecular Medicine, Berlin, Germany; [2]Friedrich Miescher Institute for Biomedical Research, Basel, Switzerland

**Abstract** Cell-fate reprograming is at the heart of development, yet very little is known about the molecular mechanisms promoting or inhibiting reprograming in intact organisms. In the *C. elegans* germline, reprograming germ cells into somatic cells requires chromatin perturbation. Here, we describe that such reprograming is facilitated by GLP-1/Notch signaling pathway. This is surprising, since this pathway is best known for maintaining undifferentiated germline stem cells/progenitors. Through a combination of genetics, tissue-specific transcriptome analysis, and functional studies of candidate genes, we uncovered a possible explanation for this unexpected role of GLP-1/Notch. We propose that GLP-1/Notch promotes reprograming by activating specific genes, silenced by the Polycomb repressive complex 2 (PRC2), and identify the conserved histone demethylase UTX-1 as a crucial GLP-1/Notch target facilitating reprograming. These findings have wide implications, ranging from development to diseases associated with abnormal Notch signaling.

*For correspondence: baris.
tursun@mdc-berlin.de (BT); rafal.
ciosk@fmi.ch (RC)

†These authors contributed
equally to this work

Present address: ‡Department
of Environmental Sciences,
Programm Mensch-Gesellschaft-
Umwelt, University of Basel,
Basel, Switzerland

Competing interests: The
authors declare that no
competing interests exist.

Reviewing editor: Julie
Ahringer, University of
Cambridge, United Kingdom

## Introduction

Cell-fate decisions are controlled, on the one hand, by intercellular signaling and, on the other hand, by intrinsic mechanisms such as epigenetic chromatin modifications. The Notch signaling pathway is a highly conserved and widespread signaling mechanism (*Artavanis-Tsakonas et al., 1999*; *Greenwald and Kovall, 2013*), which has been implicated in key cell-fate decisions such as the decision between proliferation and differentiation (*Liu et al., 2010*). Notch signaling has also been implicated in cellular reprograming. Upon inhibition of Notch signaling, the oncogenic genes KLF4 and cMyc become dispensable for the generation of induced pluripotent stem cells (iPSCs) from mouse and human keratinocytes (*Ichida et al., 2014*). In this setting, Notch inhibits reprograming. Conversely, Notch signaling promotes transdifferentiation of pancreatic acinar cells to ductal cells (*Sawey et al., 2007*), or the conversion of hepatocytes into biliary cells in liver primary malignancy intrahepatic cholangiocarcinoma (ICC) (*Sekiya and Suzuki, 2012*). Notch signaling can also affect reprograming in normal development. In *C. elegans*, signaling through the GLP-1 and LIN-12 Notch receptors impedes reprograming during embryogenesis and, during larval development, signaling through LIN-12 is required for the conversion of a rectal epithelial cell into a motorneuron (*Jarriault et al., 2008*; *Djabrayan et al., 2012*).

The role of epigenetic regulators in cell-fate decisions has been studied mostly in pluripotent cells cultured outside of their normal tissue environment (*Meshorer and Misteli, 2006*; *Spivakov and Fisher, 2007*; *Lessard and Crabtree, 2010*; *Orkin and Hochedlinger, 2011*). Therefore, the epigenetic regulation of stem cell identity in intact tissues remains poorly understood.

**eLife digest** The DNA in genes encodes the basic information needed to build an organism or control its day-to-day operations. Most cells in an organism contain the same genetic information, but different types of cell use the information differently. For example, many of the genes that are active in a muscle cell are different from those that are active in a skin cell.

These different patterns of gene activation largely determine a cell's identity and are brought about by DNA-binding proteins or chemical modifications to the DNA (which are both forms of so-called epigenetic regulation). Nevertheless, cells occasionally change their identities – a phenomenon that is referred to as reprograming. This process allows tissues to be regenerated after wounding, but, due to technical difficulties, reprograming has been often studied in isolated cells grown in a dish.

Seelk, Adrian-Kalchhauser et al. set out to understand how being surrounded by intact tissue influences reprograming. The experiments made use of *C. elegans* worms, because disturbing how this worm's DNA is packaged can trigger its cells to undergo reprograming. Seelk, Adrian-Kalchhauser et al. show that a signaling pathway that is found in many different animals enhances this kind of reprograming in *C. elegans*.

On the one hand, these findings help in understanding how epigenetic regulation can be altered by a specific tissue environment. On the other hand, the findings also suggest that abnormal signaling can result in altered epigenetic control of gene expression and lead to cells changing their identity. Indeed, increased signaling is linked to a major epigenetic mechanism seen in specific blood tumors, suggesting that the regulatory principles uncovered using this simple worm model could eventually provide insights into a human disease.

A future challenge will be to determine precisely how the studied signaling pathway interacts with the epigenetic regulator that controls reprograming. Understanding this interaction in molecular detail could help to devise strategies for controlling reprograming. These strategies could in turn lead to treatments for people with conditions that cause specific cells types to be lost, such as Alzheimer's disease or injuries.

Additionally, the impact of external cues, for example signaling from a stem cell niche to the recipient cell's chromatin remains equally unresolved. By contrast, *C. elegans* has been used as a model to study reprograming in an intact organism (*Horner et al., 1998*; *Fukushige et al., 1998*; *Zhu et al., 1998*; *Fukushige and Krause, 2005*; *Ciosk et al., 2006*; *Jarriault et al., 2008*; *Yuzyuk et al., 2009*; *Riddle et al., 2013*). In this model, germ cells can be directly reprogrammed into neurons by depleting specific chromatin modifiers such as LIN-53 (Rbbp4/7) or components of PRC2, and by concomitant overexpression of the transcription factor CHE-1, which induces glutamatergic neuronal fate in a process which we refer to as Germ cell Conversion (GeCo) (*Tursun et al., 2011*; *Patel et al., 2012*).

Here, we identify the Notch signaling pathway as a critical player in this reprograming model. This was unanticipated, since signaling through the Notch receptor GLP-1 (henceforth GLP-1[Notch]) from the somatic gonadal niche is known to maintain germline stem cell/progenitor fate (*Kimble and Crittenden, 2007*). To understand this novel, reprograming-promoting role of GLP-1[Notch], we combined genetics with tissue-specific expression profiling. We identified genes regulated by GLP-1[Notch], including genes recently shown to maintain the germline stem/progenitor cells (*Kershner et al., 2014*). Additionally, and unexpectedly, we found that many genes activated by GLP-1[Notch] signaling were also repressed by the cell fate-stabilizing chromatin regulator PRC2. We show that GLP-1[Notch] and PRC2 have an antagonistic effect on germ cell-fate decisions and demonstrate co-regulation of their common target, *utx-1*. Importantly, UTX-1 is a histone demethylase known to erase the gene-silencing methylation of histone H3 dependent on PRC2 (*Maures et al., 2011*; *Jin et al., 2011*; *Vandamme et al., 2012*). Thus, we propose that the GLP-1[Notch]–dependent induction of UTX-1 facilitates reprograming by alleviating PRC2-mediated repression of alternative cell fates.

## Results

### GLP-1$^{Notch}$ enhances conversion of germ cells into neuron-like cells

Germ cells can be converted into neuronal cells in intact *C. elegans* upon overexpression of the neuronal transcription factor CHE-1, simply by depleting the chromatin modifier LIN-53 (*Tursun et al., 2011*; *Patel et al., 2012*). This GeCo phenotype can be followed in living animals by monitoring a reporter GFP expressed from the *gcy-5* promoter, which otherwise is induced in glutamatergic ASE neurons (*Altun-Gultekin et al., 2001*). In contrast to the spontaneous teratomatous differentiation of meiotic germ cells, observed in the absence of specific RNA-binding proteins (*Ciosk et al., 2006*; *Biedermann et al., 2009*; *Tocchini et al., 2014*), GeCo is preferentially observed in the pre-meiotic, proliferating germ cells (*Tursun et al., 2011*; *Patel et al., 2012*). Consistently, removing the proliferating germ cells, by inhibiting GLP-1$^{Notch}$ signaling, prevents GeCo (*Tursun et al., 2011*). However, because the proliferating germ cells were eliminated, these experiments did not address a possible direct effect of GLP-1$^{Notch}$ signaling on GeCo. We began addressing this issue by examining the gonads of animals carrying the gain-of-function *glp-1* allele (*ar202*) (*Pepper et al., 2003*). These gonads are filled with proliferating germ cells and, upon depleting LIN-53 and overexpressing CHE-1, we observed that significantly more germ cells converted to ASE neurons (*Figure 1A*, *Figure 1—source data 1*). We refer to this enhanced GeCo as 'GeCo+'. Detailed quantification revealed that the GeCo+ gonads contained more than twice the number of converted cells (*Figure 1—figure supplement 1A*, *Figure 1—source data 1*). The nuclei of these converted cells were reminiscent of neuronal nuclei and the cells displayed axo-dendritic projection (*Figure 1—figure supplement 1B*), as previously described (*Tursun et al., 2011*; *Patel et al., 2012*). To confirm that the GeCo enhancement depends on the canonical Notch signaling pathway, rather than an independent function of the GLP-1$^{Notch}$ receptor, we RNAi-depleted the transcriptional effector of GLP-1$^{Notch}$ signaling, LAG-1 (*Christensen et al., 1996*). We exposed animals only after hatching to *lag-1* RNAi in order to avoid sterility, which is caused when animals are subjected to *lag-1* RNAi earlier (*Supplemental file 1*). RNAi-mediated knock-down of *lag-1* strongly inhibited GeCo (*Figure 1B*, *Figure 1—source data 1*). Importantly, under these experimental conditions, we did not observe any obvious reduction of germ cell numbers (*Figure 1C*, *Figure 1—source data 1*), suggesting a proliferation-independent effect of GLP-1$^{Notch}$ signaling on cell-fate conversion. To investigate this further, we tested GeCo efficiency on germ cells proliferating independently of GLP-1$^{Notch}$ signaling. We took advantage of mutants in which, in the absence of two meiosis/differentiation-promoting factors GLD-1 and GLD-2, germ cells proliferate independently of GLP-1$^{Notch}$ (*Kadyk and Kimble, 1998*). Specifically, we examined GeCo in the loss-of-function *gld-1(q497) gld-2(q485)* mutants, which carried either wild-type *glp-1* or the loss-of-function *glp-1(q175)* allele (*Austin and Kimble, 1987*). Both mutant combinations have previously been described to have tumorous germlines and impaired meiotic entry (*Kadyk and Kimble, 1998*; *Hansen et al., 2004*). In contrast to efficient GeCo observed in the *gld-1 (q497) gld-2(q485)* gonads, GeCo was strongly diminished in the *gld-1(q497) gld-2(q485); glp-1 (q175)* gonads, despite the ongoing germ cell proliferation (*Figure 1D*, *Figure 1—source data 1*). Counting the number of germ cells in these gonads revealed only a slight difference (a 15% increase in the numbers in the double vs. triple mutant gonads), suggesting that the strong enhancement of GeCo by GLP-1$^{Notch}$ signaling cannot be explained by the increased number of germ cells (*Figure 1—figure supplement 2*, *Figure 1—source data 1*). Since it has been proposed that dividing cells have a higher propensity for cellular reprograming (*Egli et al., 2008*; *Hanna et al., 2009*), we also tested whether blocking the cell cycle would affect the observed GeCo enhancement in *glp-1 (gf)* gonads. As previously described (*Fox et al., 2011*; *Patel et al., 2012*), we used hydroxyurea (HU) treatment to block the cell cycle in the S phase. Blocking the cell cycle by HU did not diminish the GeCo+ phenotype (*Figure 1—figure supplement 3*, *Figure 1—source data 1*). Combined, these results suggest that GLP-1$^{Notch}$ enhances GeCo independently from its role in promoting germ cell proliferation.

### GLP-1$^{Notch}$ activates genes mainly on X chromosomes

To understand the effects of GLP-1$^{Notch}$ on GeCo, we set out to identify genes regulated by GLP-1$^{Notch}$ signaling in germ cells. To conduct the analysis in morphologically similar tissue, we again took advantage of the *gld-1 gld-2* double mutants that, combined with either loss-of-function or

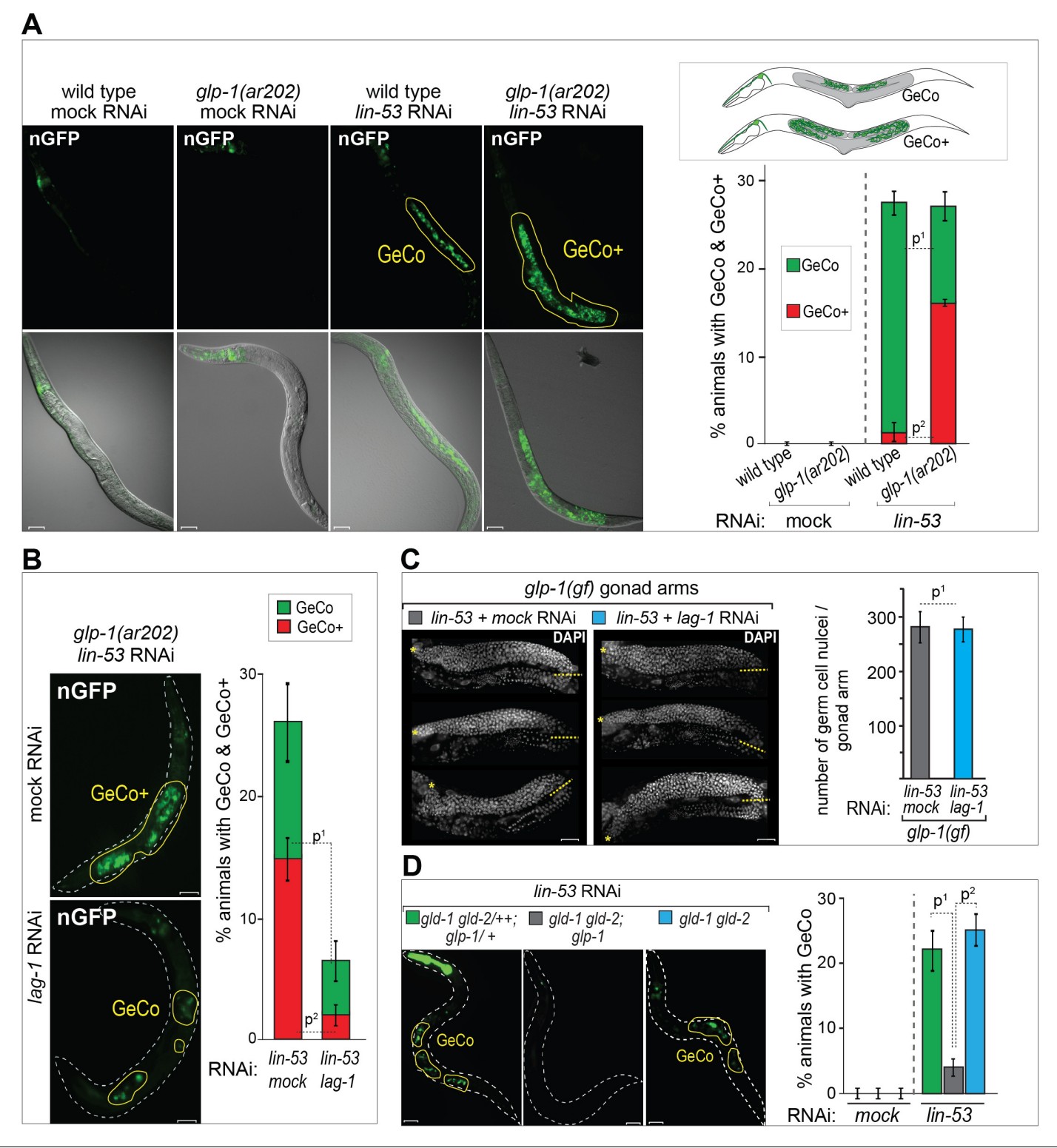

**Figure 1.** GLP-1^Notch signaling promotes reprograming of germ cells. (**A**) GLP-1^Notch enhances germ cell conversion (GeCo) into neuronal-like cells. Left: Fluorescent (top) and combined fluorescent/differential interference contrast (DIC) micrographs (bottom) of adult animals. All animals ectopically expressed the pro-neuronal transcription factor CHE-1 from a heat-shock promoter. *glp-1(ar202)* is a temperature-sensitive gain-of-function allele of the Notch receptor. Animals were subjected to either mock (control) or *lin-53* RNAi. Reprogrammed cells expressed a GFP reporter driven from the neuronal *gcy-5* promoter (here an in other figures nGFP) and are outlined here and elsewhere in yellow. Any signal outside the outlined region comes

*Figure 1 continued on next page*

*Figure 1 continued*

from somatic tissues. GeCo+ indicates animals that displayed a strongly enhanced GeCo phenotype. Scale bars = 10 µm. The cartoons depicting the GeCo and GeCo+ phenotypes are on the top right. The gonads are shaded in grey and GFP-positive converted germ cells are green. Fractions of animals displaying GeCo and GeCo+ are indicated below. At least 250 animals were quantified per condition. P-values were calculated using Student's t-test: p[1]<0,0001; p[2]=0,0006. Error bars represent SEM. (**B**) The transcriptional effector of the GLP-1[Notch] signaling pathway, LAG-1, is required for the GLP-1[Notch]–mediated enhancement of GeCo. Left: Fluorescent micrographs of adults expressing CHE-1–induced nGFP as explained above. GeCo is diminished upon the depletion of LAG-1. White dashed lines outline the animal body. Scale bars = 10 µm. Right: The corresponding quantifications. At least 400 animals were quantified per condition. P-values were calculated using Student's t-test: p[1]<0,0001; p[2]=0,0018. Error bars represent SEM. (**C**) GLP-1[Notch] signaling enhances GeCo independently from germ cell proliferation. Shown are DAPI-stained gonads of *glp-1(ar202)* animals, expressing CHE-1–induced nGFP, treated with either mock or *lin-53* RNAi. Germ cells were counted from the DTC (yellow asterisk) to the turn of the gonad arm (dashed yellow line). 15 gonad arms per condition were counted. Scale bars = 10 µm. Quantifications are on the right. While greatly inhibiting GeCo, *lag-1* RNAi did not change the number of germ cells. P-values were calculated using Student's t-test: p[1]=0,89. Error bars represent SEM. (**D**) GLP-1[Notch] enhances GeCo independently from proliferation. Left: Fluorescent micrographs of adults (with indicated genotypes), expressing CHE-1–induced nGFP. The first panel on the left shows a control, heterozygous (wild-type) *gld-1 gld-2/++; glp-1/+* animal. The other panels show the homozygous *gld-1(q497) gld-2(q485)* mutants, carrying either a loss-of-function (*q175*, center) or a wild-type (right) allele of *glp-1*. Despite proliferating, germ cells in the *gld-1 gld-2; glp-1* gonads have lost the ability to undergo GeCo. Scale bars = 10 µm. Right: the corresponding quantifications. At least 250 animals were quantified per condition. P-values were calculated using Student's t-test: p[1]=0,0478; p[2]=0,0201. Error bars represent SEM.

The following source data and figure supplements are available for figure 1:

**Source data 1.** Quantification of GeCo in *glp-1(gf)* and *lag-1* RNAi animals.

**Figure supplement 1.** *glp-1(gf)* gonads contain more than twice the number of converted cells which display neuronal characteristics.

**Figure supplement 2.** Germ cell numbers are similar between *gld-1 gld-2* double and *gld-1 gld-2; glp-1* triple mutants.

**Figure supplement 3.** Blocking the cell cycle with hydroxyurea does not inhibit GeCo+.

gain-of-function *glp-1* alleles, have morphologically similar gonads, filled with proliferating, undifferentiated germ cells (*Figure 2—figure supplement 1*) (*Kadyk and Kimble, 1998*; *Hansen et al., 2004*). We combined *gld-1(q497) gld-2(q485)* mutations with either the temperature-sensitive loss-of-function (lf) *glp-1* allele (*e2144*), or the temperature-sensitive gain-of-function (gf) *glp-1* allele (*ar202*) (*Priess et al., 1987*; *Pepper et al., 2003*). Because GLD-1 and GLD-2 regulate gene expression at the posttranscriptional level only, we expected that transcriptionally regulated GLP-1[Notch] targets could be identified in this background.

To analyze gene expression, gonads were dissected from animals grown at the restrictive temperature in two independent experiments, and transcripts were analyzed with tiling arrays (GEO accession number GSE49395). We identified around 100 transcripts that were differentially expressed between the *gld-1 gld-2; glp-1*(lf) (Notch OFF) and *gld-1 gld-2; glp-1*(gf) (Notch ON) gonads (*Figure 2A* and *Figure 2—figure supplement 2*, *Figure 2—source data 1*). These changes were confirmed by quantitative RT-PCR (RT-qPCR) on selected transcripts (*Figure 2—figure supplement2A*, *Figure 2—source data 1*). Most differentially expressed transcripts were upregulated in the 'Notch ON' gonads, indicating a predominantly activating role of GLP-1[Notch] in germ cells. For simplicity, we will refer to the transcripts upregulated at least two fold in the Notch ON gonads as 'Notch–activated'. Some Notch-activated genes, such as *sel-8/Mastermind*, *lst-1*, and *epn-1/Epsin*, have been implicated in Notch signaling in other cell types (*Doyle et al., 2000*; *Yoo, 2004*; *Tian et al., 2004*; *Singh et al., 2011*; *Kershner et al., 2014*). However, it remains possible that, rather than being direct GLP-1[Notch] targets, some of the Notch-activated genes were upregulated as an indirect consequence of increased GLP-1[Notch] signaling.

To demonstrate that Notch-activated genes are functionally relevant for germ cell proliferation, we performed RNAi knockdown of Notch-activated genes (n = 64) on animals carrying the gain-of-function *glp-1(ar202)* allele, and screened for enhancement or suppression of the tumorous gonad phenotype (*Supplementary file 1*; for detailed experimental procedure see Materials and Methods). Knocking down some of the Notch-activated genes suppressed the tumorous phenotype, which agrees with predominantly proliferation-promoting role of GLP-1[Notch]. Interestingly, knocking down a smaller subset of the Notch-activated genes enhanced the tumor (*Supplementary file 1*),

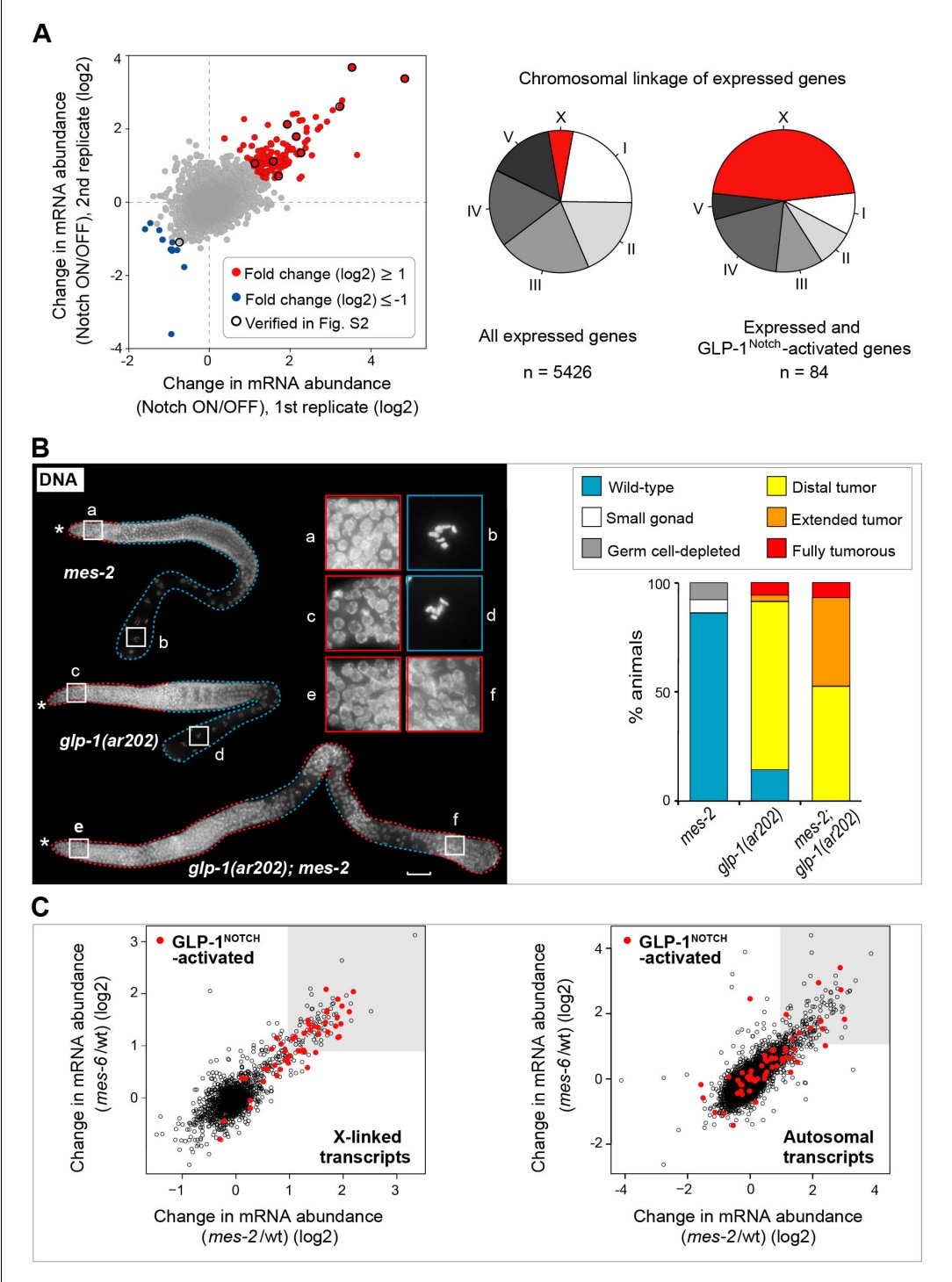

**Figure 2.** GLP-1[Notch] and PRC2 regulate common targets and are functionally connected. (**A**) Notch-activated genes are biased for the sex chromosome linkage. Left: Changes in transcript abundance in the 'Notch ON' versus 'OFF' dissected gonads (genotypes explained in *Figure 2— figure supplement 1A–B*) were analyzed by microarrays. Transcripts upregulated at least 2-fold in the 'Notch ON' gonads are marked in red, those downregulated at least 2-fold in blue. Selected transcripts verified by RT-qPCR in *Figure 2—figure supplement 2A* are additionally circled in black. Right: 5426 genes can be considered expressed in the gonad, based on the bimodal distribution of expression values. Only 3% of those expressed genes are X-linked. In contrast, nearly half (46%) of the expressed and Notch-activated transcripts are X-linked (see *Figure 2—figure supplement 2B* for numbers). (**B**) GLP-1[Notch] and PRC2 interact genetically. Left: DAPI-stained gonads from animals of the indicated genotypes. The *mes-2(bn11)* M+Z- single mutant gonads have wild-type appearance at 20°C. The *glp-1(ar202)* gain-of-function mutants have an almost wild-type appearance at this

*Figure 2 continued on next page*

*Figure 2 continued*

temperature, except for an extended proliferative zone in the gonad, referred to as 'distal tumor'. At the same temperature, *mes-2(bn11)* M+Z-; *glp-1 (ar202)* double mutants developed germline tumors in 32/32 of the examined gonads. The insets show close-ups from the indicated gonadal regions: the distal-most regions contain undifferentiated, proliferative germ cells in all mutants (a, c, e). However, while the single mutants contain oocytes with characteristically condensed chromosomes in the proximal gonads (b, d), the proximal gonads of the double mutants harbor proliferative germ cells (f). Scale bar = 30 μm. Right: quantification of the phenotypes. 'Distal tumor' indicates the presence of an elongated distal proliferative zone (approximately ½ of the distal gonad arm). 'Extended' tumor indicates an extended distal tumor, few oocytes, and frequently also a proximal tumor. 'Fully tumorous' indicates the absence of all differentiated cell types except for sperm produced during larval development. (C) GLP-1$^{Notch}$ and PRC2 target the same genes on the X chromosomes. The plots correlate changes in gene expression in M+Z- *mes-2* mutants with changes in gene expression changes in M+Z- *mes-6* mutants. Results are shown separately for X-linked (left) and autosomal (right) transcripts. Notch-activated genes (red in *Figure 2A*) are marked in red. Lightly shaded areas indicate transcripts that are at least 2-fold upregulated. The overlap between transcripts upregulated by GLP-1$^{Notch}$ and transcripts upregulated by the loss of CePRC2 is highly significant, particularly for the X-linked genes. The significance of the correlation was measured by hypergeometric distribution; X-linked Notch-activated vs. *mes-2* derepressed: p=1.31e-31; X-linked Notch-activated vs. *mes-6* de-repressed: p=7.41e-25; autosomal Notch-activated vs. *mes-2* derepressed: p=1.47e-22; autosomal Notch-activated vs. *mes-6* de-repressed: p=1.8e-12.

The following source data and figure supplements are available for figure 2:

**Source data 1.** Microarray results.

**Figure supplement 1.** Examining transcriptional effects of GLP-1$^{Notch}$ signaling.

**Figure supplement 2.** Analysis of Notch-activated genes.

**Figure supplement 3.** The PRC2 component MES-6 and most enhancers/suppressors of *glp-1(ar202)* induced tumors appear to interact genetically with GLP-1$^{Notch}$ signaling in a germline-autonomous manner.

**Figure supplement 4.** Global levels of H3K27me3 are unaffected by neither loss-of-function nor gain-of-function mutations in *glp-1*.

suggesting that some of the Notch-activated genes may counteract proliferation. While some of these genes may function autonomously in the germline, others could affect the germline indirectly from the soma. To test this, we RNAi-depleted selected candidates in the *rrf-1 (pk1417)* mutant background, which is permissive for RNAi in the germline but deficient in RNAi in many (but not all) somatic tissues (*Kumsta and Hansen, 2012*). While depleting most candidates in the *rrf-1* background had similar effects on the germline as in the wild type (suggesting germline-autonomous function), in some cases the effects were abolished, suggesting that these genes function in the soma (*Figure 2—figure supplement 3*, *Figure 2—source data 1*).

Strikingly, we noticed that Notch-activated genes were enriched on the X-chromosome, the *C. elegans* sex chromosome. 45% of the Notch-activated genes were X-linked, which is four-fold more than expected by chance (p=2.99e$^{-14}$; *Figure 2A* and *Figure 2—figure supplement 2B*, *Figure 2— source data 1*). When analyzing only genes with higher than baseline germline expression values, the disproportional X-linkage of Notch-activated genes was even more striking (fifteen times more than expected by chance (p=2,19e$^{-38}$; *Figure 2A* and *Figure 2—figure supplement 2B*, *Figure 2— source data 1*). In the *C. elegans* germline, X-linked genes are largely silenced by the *C. elegans* PRC2 (*Fong et al., 2002*). Thus, the X chromosome bias among Notch-activated genes suggested a possible antagonistic relationship between GLP-1$^{Notch}$ and PRC2.

## GLP-1$^{Notch}$ and PRC2 have antagonistic functions in the germline

The *C. elegans* PRC2 consists of MES-2, -3, and -6 (*Bender et al., 2004*) and levels of the corresponding transcripts were essentially not altered by GLP-1$^{Notch}$ signaling (*mes-2*: absolute fold change (fc) -1.3747; *mes-3*: fc 1.003; *mes-6*: fc 1.037). To test for a functional relationship between GLP-1$^{Notch}$ and PRC2, we examined genetic interactions between GLP-1 and PRC2 mutants. At 20°C, both the temperature-sensitive gain-of-function *glp-1(ar202)* and the homozygous loss-of-function *mes-2(bn11)* mutants, derived from heterozygous mothers providing maternal MES-2 (*mes-2* M+Z- mutants), were viable and produced gonads with nearly wild-type appearance. The double *mes-2* M+Z-; *glp-1(ar202)* mutants, however, displayed distal and proximal tumors at the same

temperature (*Figure 2B*; 32/32 examined gonads). PRC2 and GLP-1$^{Notch}$ thus interact functionally, and they appear to do so in a germ cell autonomous manner (*Figure 2—figure supplement 3*, *Figure 2—source data 1*).

Given the striking enrichment of Notch-activated genes on the X chromosome, and the genetic interaction between PRC2 and GLP-1$^{Notch}$, we speculated that GLP-1$^{Notch}$ and PRC2 act on similar genes. To determine whether Notch-activated genes are also PRC2-repressed, we first determined putative PRC2 targets by expression analyses on isolated wild-type, M+Z- *mes-2* or *mes-6* mutant gonads (GEO accession number GSE49395). Consistent with the joint function of MES-2 and MES-6 in the PRC2 complex, a very similar set of genes was upregulated upon the loss of either protein (*Figure 2C*; *Figure 2—source data 1*; henceforth 'PRC2 repressed' genes; also see (*Gaydos et al., 2012*). Indeed, those PRC2-repressed genes overlapped strongly with Notch-activated genes, particularly those linked to the X chromosome. Nearly all of the X-linked Notch-activated genes were also derepressed upon the loss of PRC2 (*Figure 2C*). This is consistent with the observed genetic interaction and suggests that increased GLP-1$^{Notch}$ signaling can induce expression of specific PRC2-repressed genes. This activation of the PRC2-repressed genes is not due to a global loss of the repressive tri-methylation of Histone H3 at lysine residue 27 (H3K27me3), since, upon examining gonads of different GLP-1$^{Notch}$ mutants, we observed no global loss of H3K27me3 in the germline (*Figure 2—figure supplement 4*).

## GLP-1$^{Notch}$ signaling enhances reprograming

Germ cell conversion to neurons can be triggered not only by LIN-53 depletion but also by the depletion of PRC2 (*Patel et al., 2012*). Potentially, the depletion of LIN-53 could facilitate reprograming by inhibiting PRC2, since the depletion of LIN-53 results in a global loss of H3K27me3 in the germline (*Patel et al., 2012*). Considering the antagonistic relation between GLP-1$^{Notch}$ and PRC2 on cell proliferation and gene regulation, we wondered whether GeCo triggered by the depletion of PRC2 would be sensitive to Notch signaling. Indeed, GeCo was strongly enhanced in PRC2-depleted (*mes-2*, *mes-3* or *mes-6* RNAi) animals, when they also carried the gain-of-function *glp-1(ar202)* allele (*Figure 3A*, *Figure 3—source data 1*). Moreover, similar to the LIN-53–depleted *gld-1(q497) gld-2 (q485); glp-1(q175)* gonads (*Figure 1D*), the loss of GLP-1 effectively prevented GeCo in PRC2–depleted *gld-1(q497) gld-2(q485); glp-1(q175)* gonads (*Figure 3B*, *Figure 3—source data 1*). Together, these results suggest that GLP-1$^{Notch}$ stimulates GeCo in PRC2-compromised gonads.

## UTX-1 is required for GLP-1$^{Notch}$-mediated GeCo enhancement

To determine how GLP-1$^{Notch}$ might counteract PRC2, we depleted candidate Notch-activated genes (*Supplementary file 1*), and examined GeCo efficiency (*Figure 4A*, *Figure 4—source data 1*). The strongest suppression of the GeCo+ and GeCo phenotype was observed upon the depletion of *utx-1*, which also suppressed *mes-3* RNAi-mediated GeCo+ in *glp-1(ar202)* gonads (*Figure 4—figure supplement 1*). Depletion of two other candidates, the uncharacterized C07G1.6 and the aldolase ortholog *aldo-1*, also suppressed GeCo+, albeit less efficiently (*Figure 4A*). Because UTX-1 was suggested to effect gonadal development by functioning in the somatic gonad (*Vandamme et al., 2012*), we re-examined GeCo efficiency upon *utx-1* RNAi in the *rrf-1 (pk1417)* background. Importantly, in *rrf-1* mutants, RNAi is impaired in the somatic gonad, including the distal tip cell (DTC), which constitutes the germline stem cell niche (*Kumsta and Hansen, 2012*). Because the suppression of GeCo+ upon *utx-1* RNAi was observed also in the *rrf-1* background, UTX-1 does not seem to enhance GeCo by functioning in the somatic gonad (*Figure 4—figure supplement 1A*). Furthermore, different genetic backgrounds and RNAi agains *utx-1* do not affect the expression levels of CHE-1 in the germline (*Figure 4—figure supplement 1B*).

Importantly, *utx-1* encodes a conserved H3K27me3 demethylase, an enzyme erasing the repressive mark deposited by PRC2 (*Agger et al., 2007*; *Jin et al., 2011*; *Maures et al., 2011*), potentially explaining how its depletion impairs GeCo efficiency. However, a number of other H3K27me3 demethylases exist in *C. elegans,* which prompted us to test whether depletion of these demethylases might have an effect on GeCo in the *glp-1(ar202)* gonads. We RNAi-depleted *jmjd-1.2*, encoding a H3K9/27me2 demethylase (*Kleine-Kohlbrecher et al., 2010*), *jmjd-3.1*, *jmjd-3.2*, and *jmjd-3.3*, which were reported to demethylate H3K27me2/3 (*Agger et al., 2007*; *Kleine-Kohlbrecher et al., 2010*; *Zuryn et al., 2014*), and, as a control, *jmjd-2,* encoding a H3K9/36 demethylase (*Whetstine et al.,*

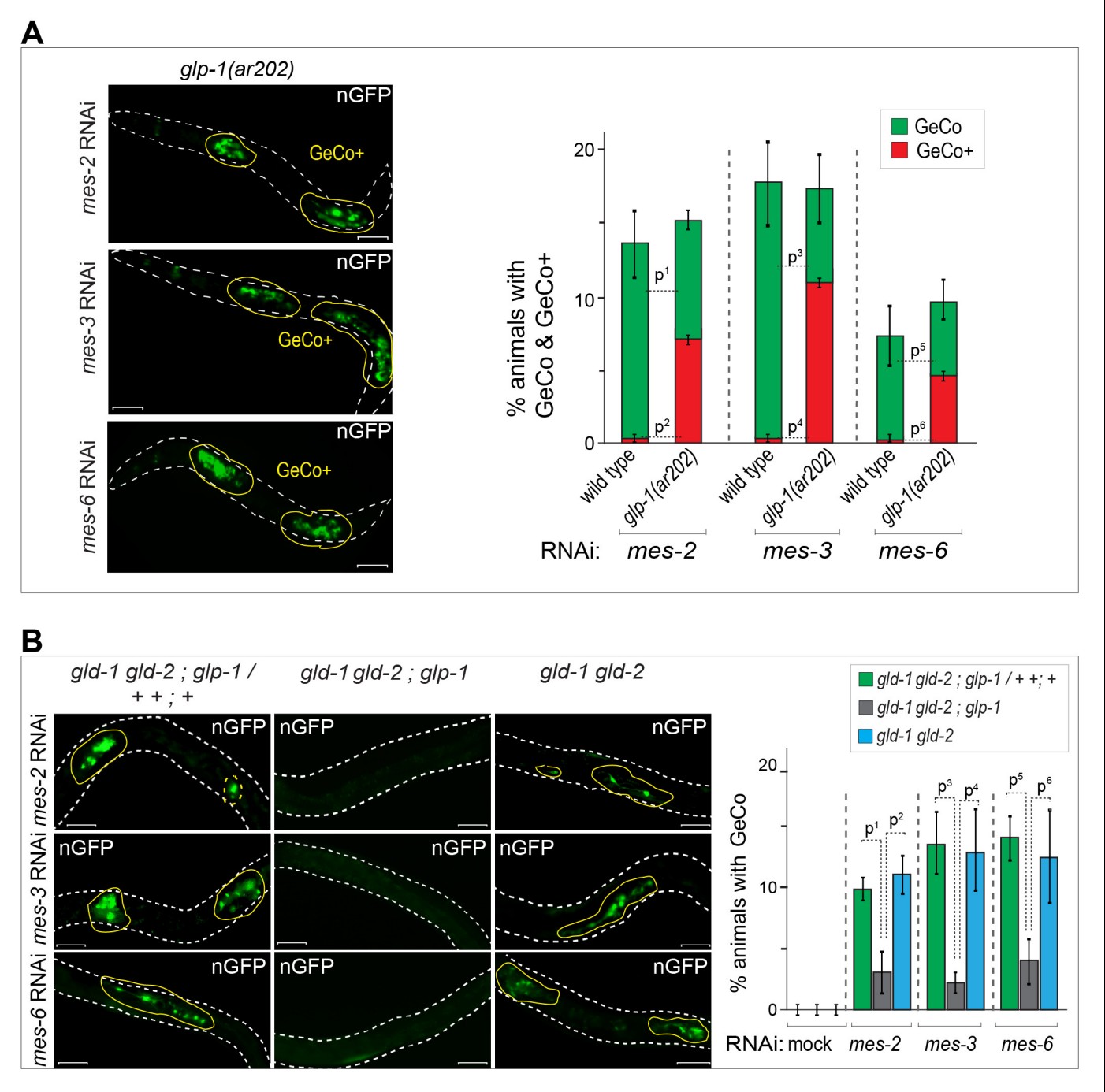

**Figure 3.** GLP-1[Notch] enhances reprograming upon the depletion of PRC2. (**A**) Notch and PRC2 genetically interact in GeCo. Left: Fluorescent micrographs of *glp-1(ar202)* gain-of-function mutants expressing CHE-1–induced neuronal GFP. The animals were subjected to control RNAi or RNAi against PRC2 components (MES-2, 3, and 6), as indicated. Increased GLP-1[Notch] signaling enhanced the GeCo+ phenotype upon PRC2 depletion. Control RNAi (mock) for each genetic background did not result in any GeCo (images not shown – see quantification). Right: The corresponding quantifications. P-values were calculated using Student's t-test: $p^1$=0,0006; $p^2$<0,0001; $p^3$=0,0536; $p^4$=0,0001; $p^5$=0,4035; $p^6$=0,0003. At least 200 animals were scored per condition. Error bars represent SEM. (**B**) GLP-1[Notch] is required for GeCo in PRC2-depleted animals independently from proliferation. Left: Fluorescent micrographs of adults expressing CHE-1–induced nGFP, with the genotypes indicated above the panels. The animals were subjected to RNAi as indicated on the left. The first column shows heterozygous, the other two homozygous animals carrying the loss of function alleles *gld-1(q497)*, *gld-2(q485)* and, in the central panels, *glp-1(q175)*. The animals were subjected to control RNAi or RNAi against PRC2 components (MES-2, 3, and 6). In the absence of GLP-1[Notch], depletion of PRC2 components did not induce GeCo. Scale bars = 10 μm. Right: The corresponding

*Figure 3 continued on next page*

*Figure 3 continued*

quantifications. P-values were calculated using Student's t-test: $p^1 < 0.0456$; $p^2 = 0.0337$; $p^3 = 0.0070$; $p^4 = 0.0637$; $p^5 = 0.0080$; $p^6 = 0.1259$. At least 70 animals were scored per condition. Error bars represent SEM.
The following source data is available for figure 3:

**Source data 1.** Quantification of GeCo upon PRC2 depletion.

*2006*; *Greer et al., 2014*) (*Figure 4B*, *Figure 4—source data 1*). Only the depletion of JMJD-1.2 suppressed GeCo+ significantly, though to a lesser degree than the depletion of UTX-1 (*Figure 4B*, *Figure 4—source data 1*). The suppression of GeCo by the depletion of UTX-1 or JMJD-1.2 stresses the importance of counteracting PRC2 in reprograming. However, only the expression of *utx-1* is activated by the GLP-1[Notch] signaling, suggesting that it is the activity of UTX-1 which is key for the reprograming dependent on GLP-1[Notch] signaling.

## GLP-1[Notch] and PRC2 regulate expression of the H3K27 demethylase UTX-1

The inhibition of GeCo enhancement upon *utx-1* RNAi in the *rrf-1* background suggested that UTX-1 functions in the germline. To test the potential expression of *utx-1* in the germline, we constructed a strain expressing single copy-integrated, FLAG and GFP-tagged, functional UTX-1 (expressed from the endogenous *utx-1* promoter under the control of endogenous *utx-1* 3'UTR). We found that this protein was indeed expressed in the germline (*Figure 5—figure supplement 1*). To examine the potential transcriptional regulation of *utx-1* expression by GLP-1[Notch] and PRC2, we also created a strain expressing GFP-tagged histone H2B from the *utx-1* promoter (*putx-1* reporter), under the control of an unregulated (*tbb-2*) 3'UTR. This construct was also single-copy integrated into a defined genomic location. With both the UTX-1 fusion protein and the *putx-1* reporter, we expected expression patterns similar to that of other reported GLP-1[Notch] target genes (*Lamont et al., 2004*; *Lee et al., 2006*; *Kershner et al., 2014*). Among these, *lst-1* and *sygl-1* account for the role of GLP-1[Notch] in stem cell maintenance, and both genes are expressed in the distal-most stem cells/progenitors (*Kershner et al., 2014*). By contrast, both the UTX-1 fusion protein and the *putx-1* reporter were little or not expressed in the distal-most, proliferative part of the germline (*Figure 5A–C*; *Figure 5—figure supplement 1*). Concomitantly with progression through meiosis, their expression increased toward the proximal end of the gonad (*Figure 5A–C*). When examining the existing mRNA hybridization patterns of Notch-activated and PRC2-repressed genes (33 genes), we noticed that half of these (18, all X-linked) are similarly expressed in the medial and/or proximal, but not the distal-most, gonads (*Supplementary file 2*), arguing against direct transcriptional activation of these genes by GLP-1[Notch] in the wild type. Nevertheless, in agreement with the expression analyses, we observed increased expression of the *putx-1* reporter in PRC2-depleted gonads; this increase occurred throughout the gonad, including the distal-most region (*Figure 5A*). In situ hybridization for the endogenous *utx-1* transcript also showed increased expression throughout the gonad in the absence of PRC2 (*Figure 5—figure supplement 2*). Moreover, expression of the *putx-1* reporter was higher upon increased GLP-1[Notch] activity in *glp-1(ar202)* mutants, including the distal-most region where, in the wild type, *utx-1* is little or not expressed (*Figure 5B*). Importantly, we found that the activation of the *utx-1* promoter by Notch signaling was depended on the putative LAG-1/CSL binding sites within the promoter (*Yoo et al., 2004*), as deleting those sites reduced reporter expression by approximately one-fourth (*Figure 5C*). The interaction between LAG-1 and *utx-1* was also tested by chromatin immunoprecipitation (ChIP), performed on a strain expressing FLAG-tagged LAG-1 in either wild-type or *glp-1(ar202)* background (*Figure 5D* and *Figure 5—figure supplement 3*); the previously identified GLP-1[Notch] targets, *lst-1* and *sygl-1* (*Kershner et al., 2014*), served as positive controls for the ChIP. Expectedly, we observed the enhanced binding of LAG-1 to the *utx-1* promoter, indicating that *utx-1* is a transcriptional target of GLP-1[Notch] signaling. Summarizing, based on the observations in mutant backgrounds, PRC2 and GLP-1[Notch] signaling have antagonistic effects on *utx-1* transcription. However, in wild type, the endogenous levels of GLP-1[Notch] signaling are apparently insufficient to overcome PRC2-mediated repression of *utx-1* in the distal-most part of the gonad.

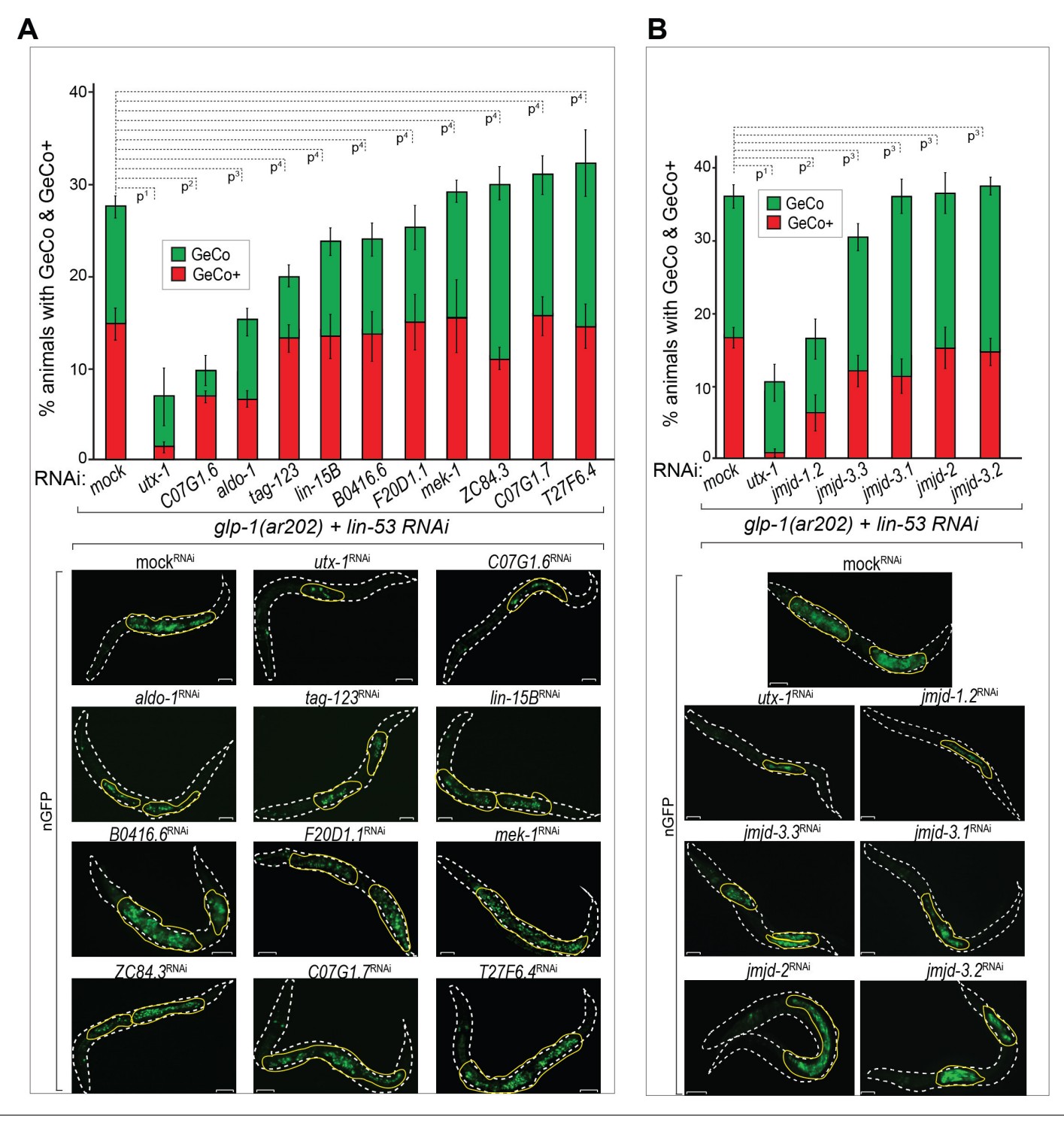

**Figure 4.** The H3K27 demethylase UTX-1 is required for GeCo enhancement. (**A**) UTX-1 is critical for GeCo enhancement. Candidate Notch-activated genes, selected from *Supplementary file 2* with available RNAi clones, were assayed for a role in GeCo in *glp-1(ar202)* animals, expressing CHE-1–induced nGFP and treated with *lin-53* RNAi. While the additional depletion of *utx-1* had the strongest impact on GeCo+ and GeCo, the depletion of C07G1.6 and *aldo-1* had a weaker effect. Representative fluorescence micrographs are below the quantification chart. White dashed line outline the animal body, yellow lines outline gonadal areas with GeCo. P-values for GeCo+ were calculated using Student's t-test: $p^1$=0,000013; $p^2$=0,026; $p^3$=0,021; $p^4$>0,1. At least 250 animals were scored per condition. Error bars represent SEM. nGFP = *gcy-5::gfp*. Scale bars = 10 μm. (**B**) As in A, but RNAi was performed against *jmjd-1.2* (H3K9/27me2 demethylase); *jmjd-3.1, jmjd-3.2,* and *jmjd-3.3,* (H3K27me2/3 demethylases); and *jmjd-2* (H3K9/36 demethylase). Only RNAi against *jmjd-1.2* suppresses GeCo+, though to a lesser degree compared to *utx-1* RNAi. Representative fluorescence

*Figure 4 continued on next page*

*Figure 4 continued*

micrographs are below the quantification chart. P-values for GeCo+ were calculated using Student's t-test: $p^1$=0,0042; $p^2$=0,035; $p^3$>0,2. At least 190 animals were scored per condition. Error bars represent SEM. Scale bars = 10 µm.

The following source data and figure supplement are available for figure 4:

**Source data 1.** Quantification of GeCo upon double RNAi against *lin-53* and Notch-activated genes.

**Figure supplement 1.** UTX-1 is required for the GeCo+ enhancement upon the depletion of PRC2.

## Discussion

In the *C. elegans* germline, GLP-1[Notch] signaling is essential for maintaining a pool of undifferentiated stem cells/progenitors. Here, we report an unexpected role of GLP-1[Notch] signaling in promoting cell fate reprograming. To understand this phenomenon, we identified genes activated upon increased GLP-1[Notch] signaling. While the identified genes include the physiological GLP-1[Notch] targets, *sygl-1* and *lst-1*, many other genes, including *utx-1*, appear to be only weakly or not expressed in the distal-most region of the wild-type gonad, where both *sygl-1* and *lst-1* are induced by GLP-1[Notch] (*Kershner et al., 2014*). Thus, the wild-type levels of GLP-1[Notch] signaling are either insufficient to induce expression of many potential target genes (see below), or their expression is controlled by additional mechanisms, perhaps similarly to *lip-1* mRNA, which, while induced by GLP-1[Notch], is post-transcriptionally degraded in the distal-most gonad (*Hajnal and Berset, 2002*; *Lee et al., 2006*). In addition to its major proliferation-promoting function, GLP-1[Notch] has been suggested to have a lesser role in promoting germ cell differentiation (*Hansen et al., 2004*). Some of the identified Notch-activated genes appear to promote germ cell differentiation, potentially explaining the proposed differentiation-promoting function of GLP-1[Notch]. However, whether these genes are activated by GLP-1[Notch] and promote differentiation under physiological conditions remains to be determined.

Many of the Notch-activated genes are repressed by PRC2, suggesting that the expression of these genes is determined by the crosstalk between the extrinsic intercellular signaling pathway, Notch, and the intrinsic chromatin modifier PRC2. Indeed, at least in the case of *utx-1*, PRC2 appears to prevent its inappropriate expression in the distal-most gonad, which, nevertheless, can be overcome upon increased GLP-1[Notch] signaling. Our findings suggest that GLP-1[Notch] antagonizes PRC2, at least in part, by stimulating expression of the H3K27 demethylase UTX-1, which is essential for the enhancement of cellular reprograming. Previously, it was suggested that UTX-1 influences the germline by functioning in the somatic gonad (*Vandamme et al., 2012*). However, by using the *rrf-1* background, which displays defective RNAi in the somatic gonad, including the DTC (*Kumsta and Hansen, 2012*), we found that the reprograming-promoting role of UTX-1 is unlikely to depend on its function in the somatic gonad. Although we cannot fully exclude the possibility that the reprograming-facilitating role of UTX-1 depends on its expression in another somatic tissue (such as the intestine, in which RNAi remains functional in the *rrf-1* mutant; *Kumsta and Hansen, 2012*), the germline expression of UTX-1 reported here suggests a germline-autonomous function. Consistent with this hypothesis, manipulating either PRC2 or GLP-1[Notch] affected the germline expression of *utx-1*.

Although additional factors, such as the uncharacterized C07G1.6 and the ortholog of the human aldolase A (*Kuwabara and O'Neil, 2001*; *Shaye and Greenwald, 2011*) might contribute to reprograming, they appear to be less critical. In addition to UTX-1, another H3K27/H3K9-demethylating enzyme, JMJD-1.2 (*Kleine-Kohlbrecher et al., 2010*), is required for enhanced reprograming. Similar to UTX-1, JMJD-1.2 is likely to be directly involved in regulating chromatin accessibility, since its depletion results in increased levels of the repressive H3K9me2 and H3K27me2 modifications (*Kleine-Kohlbrecher et al., 2010*). The reprograming-promoting role of JMJD-1.2 might indicate that, besides further demethylation of H3K27me2, perhaps also the removal of H3K9me2 facilitates GeCo. Future studies will shed light on this interesting question.

UTX-1 orthologs in other species contribute to tissue-specific chromatin signatures, for example during myogenesis or in cardiac development (*Seenundun et al., 2010*; *Lee et al., 2012*), and have

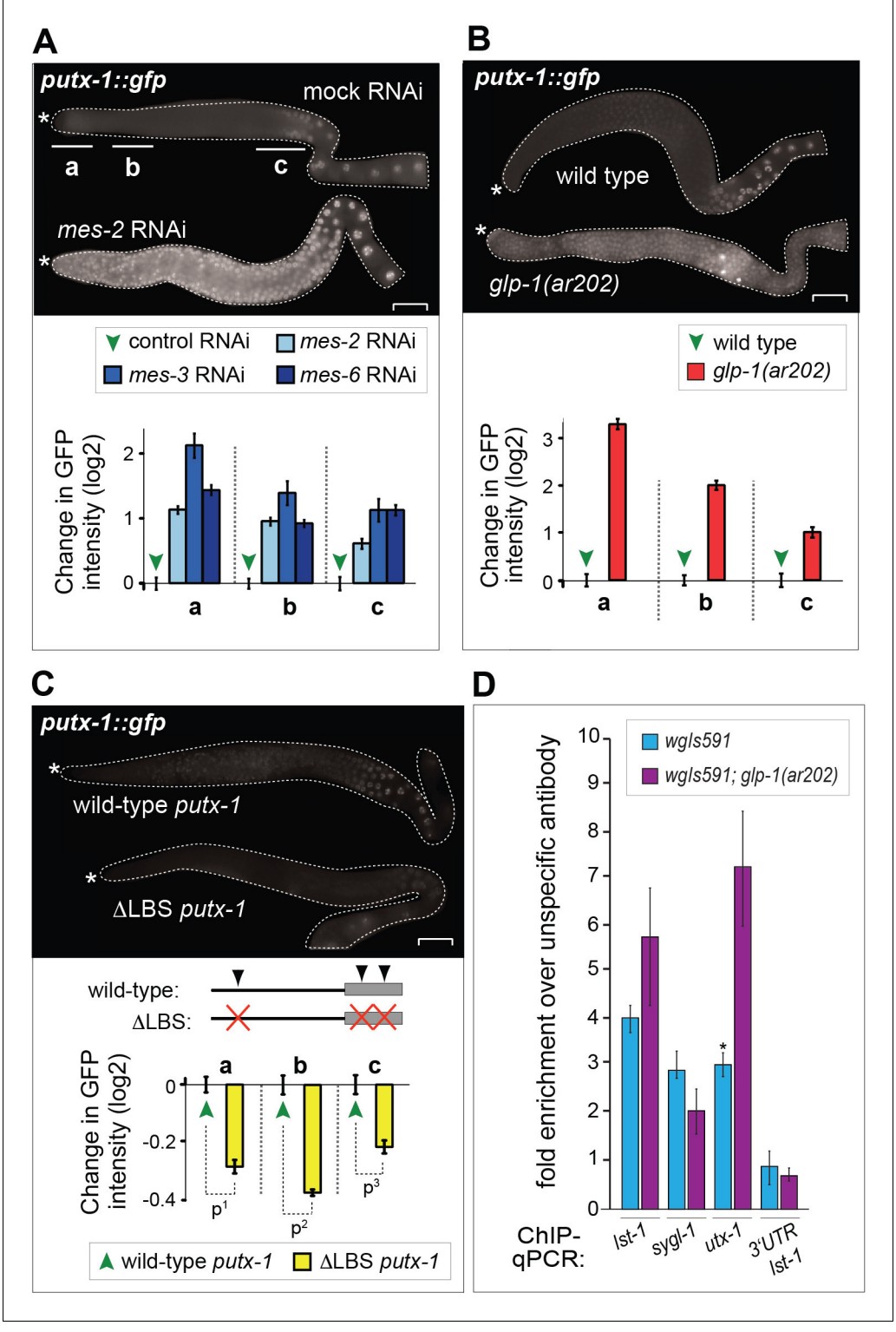

**Figure 5.** UTX-1 is regulated by GLP-1[Notch] and PRC2. (**A–C**) Expression of *utx-1* is regulated by PRC2 and GLP-1[Notch]. Top: dissected gonads expressing a GFP reporter, driven from the *utx-1* promoter (*putx-1::GFP*, fused to histone 2B for nuclear localization to facilitate quantification), subjected to the indicated RNAi (**A**), crossed into the indicated genetic background (**B**) or carrying the indicated mutations in the reporter gene (**C**). a, b, and c indicate gonadal regions containing germ cells in mitosis (a), and leptotene/zygotene (b) or pachytene (c) stages of meiosis. Below: the corresponding GFP quantifications. The diagrams show GFP intensities relative to the control (indicated by green arrows) in regions a-c. Results are represented as average changes in the GFP intensity

*Figure 5 continued on next page*

*Figure 5 continued*

(relative to mock RNAi-ed or untreated animals). The error bars represent SEM. The numbers of analyzed gonads were as follows: n = 44 for wild-type reporter; n = 36 for *glp-1(ar202)*; n = 55 for wild-type reporter on control RNAi; n = 48 for *mes-2* (RNAi); n = 15 for *mes-3* (RNAi); n = 29 for *mes-6* (RNAi), and n = 20 for the LAG-1 binding sites deleted reporter. (A) The *putx-1::GFP* reporter is repressed by PRC2. In all *mes-depleted* gonads, the reporter was de-repressed in proliferating cells (a) as well as in more proximal gonadal regions (b-c). (B) The *putx-1::GFP* reporter is upregulated by GLP-1[Notch]. In the gain-of-function *glp-1(ar202)* mutant, the reporter was strongly derepressed in the proliferating cells in the distal-most gonad (a). Its expression was also increased in the more proximal regions (b-c), which, in this mutant background, contain proliferating cells instead of meiotic cells. (C) *putx-1::GFP* expression depends on the predicted LAG-1/CSL binding sites in the promoter. Upon deletion of putative LAG-1/CSL binding sites, the reporter expression was abolished. The changes in GFP intensities were highly significant (p-values were measured by independent t-tests) $p^1$=4.85$^{-13}$, $p^2$=1.38$^{-20}$, $p^3$=1.18$^{-7}$. (D) LAG-1 binds the *utx-1* promoter. Lysates of animals expressing FLAG-tagged LAG-1 (strain *wgIs591; lag-1::TY1:: EGFP::3xFLAG*), either in wild-type or *glp-1(ar202)* background, were subjected to ChIP-qPCR analysis of the indicated genes. Negative controls and additional tested genes are shown in *Figure 5—figure supplement 3*. The qPCR amplicons were tested in at least three independent experiments. The results are shown as fold enrichment in anti-FLAG IP compared to IP with unspecific antibody. The 3'UTR of *lst-1* serves as a negative control. Interestingly, LAG-1 binding in the *glp-1(ar202)* gain-of-function background is stronger to the *utx-1* promoter than to the reported GLP-1[Notch] targets *lst-1* and *sygl-1*. The asterisk indicates a p-value < 0.05 (Students t-test). Error bars represent SEM.

The following figure supplements are available for figure 5:

**Figure supplement 1.** The functional *utx-1* transgene is expressed in the same pattern as a GFP reporter coupled to the *utx-1* promoter.

**Figure supplement 2.** The endogenous *utx-1* mRNA is upregulated in the absence of the PRC2-component MES-6.

**Figure supplement 3.** Testing LAG-1 binding to additional genes by ChIP.

---

been implicated in germ-cell and somatic reprograming (*Mansour et al., 2012*). Together with our data, these findings underscore the importance of UTX-1 in cellular reprograming. Here, we suggest that one way the activity of UTX-1 may be stimulated to promote reprograming is through its Notch signaling-dependent transcriptional activation. Interestingly, an antagonistic relationship between Notch and PRC2 has also been observed in T-cell leukemia (*Ntziachristos et al., 2012*). A fascinating possibility is that a regulatory principle described here could help explain the etiology of this and perhaps other human diseases linked to a pathological increase in Notch signaling.

## Materials and methods

### Nematode culture and mutants

Standard procedures were used to maintain animals. Worms (RRID:WB_DL226) were grown at 25°C unless stated otherwise. All heat-shock and temperature-sensitive strains were kept at 15°C. The *C. elegans* lines used in this study are listed and described in detail in the *Supplementary file 3A*.

### RNA interference experiments

The enhancer-suppressor screen on Notch targets was performed by feeding the animals with bacteria containing RNAi clones from the Ahringer and Vidal RNAi libraries. The clones not present in either of these libraries were cloned using primers as described in detail in the *Supplementary file 3B*. Experiments were performed at 15°C, 20°C or 25°C using bleached embryos or overnight-synchronized L1 animals as stated in *Supplementary file 1*. The percentage of adult animals with the germline tumor phenotype was scored. To test germ-cell autonomy, RNAi clones that induced significant suppression or enhancement in either setting were re-tested in a strain carrying the *glp-1 (ar202)* (RRID:WB_GC833) allele and, additionally the *rrf-1(pk1417)* (RRID:WB_NL2098) mutation, which impairs RNAi primarily in the soma (*Sijen et al., 2001*). For this test, gravid adults were picked

to RNAi plates and allowed to lay eggs overnight at the semi-permissive temperature of 20°C. Progeny was scored for enhancement or suppression of the germline tumorous phenotype at the young adult stage by DAPI staining of dissected gonads and scoring gonads as either wild-type, as containing a proximal or distal tumor but still some eggs, or as completely tumorous.

Reprograming experiments were carried out as F1-RNAi. Worms were put on RNAi plates and the following F1 generation was screened for phenotypes. Used RNAi clones are described in the *Supplementary file 3B*. The genotype of the worms used for germ cell reprograming assays is BAT28 (*otls305 ntls1,* RRID:WB_OH9846, details in Supplemental Materials and methods). Animals were synchronized by bleaching and eggs were put on NGM-agar containing *E. coli* OP50 (RRID: WB_OP50) as a food source to grow at 15°C until worms reached the L4-stage. At this stage 15–20 worms were put per well of a 6-well plate, containing bacteria expressing dsRNA or carrying an empty RNAi vector, and grown at 15°C until most of the F1 progeny reached the L3/L4 stage. The plates were heat-shocked at 37°C for 30 min followed by an overnight incubation at 25°C. Next day (~16 hrs post heat-shock) the plates were screened for progeny showing ectopic GFP induction in the germline. To induce the Glp phenotype in *glp-1(ar202)*, the animals were shifted to room temperature 9 hrs before the heat-shock. For double RNAi, bacteria were grown as saturated culture. The $OD_{600}$ was measured to ensure that the bacteria were mixed in an appropriate 1:1 ratio and subsequently seeded on RNAi 6-well plates. The RNAi screen was performed as described above. The p-values were calculated using Student's *t*-test.

## Cell cycle arrest by HU treatment and EdU staining

Hydroxyurea (HU) treatment was carried out as previously described (*Fox et al., 2011*; *Patel et al., 2012*). HU was added to seeded RNAi-plates at a final concentration of 250 μM. L4 worms (strain BAT32, details in *Supplementary file 3A*) grown on RNAi-plates were transferred to HU plates and incubated at room temperature for 5 hrs prior to heat-shock in order to induce CHE-1 expression. To test HU efficiency, control animals were treated with HU for 12 hrs with subsequent staining for DAPI and H3Ser10ph (pH3) antibody (Abcam #ab5176). After overnight incubation, the worms were assessed for GFP induction in the germline as described above. To assess the efficiency of the HU treatment, the *E. coli* strain MG1693 (*E. coli* stock center), with incorporated 5-ethynyl-20-deoxyuridine (EdU), was used to feed L3/L4 worms. EdU in combination with DAPI staining was performed similar to the procedure described previously (*Ito and McGhee, 1987*) and according to the manufacturer's instructions (Invitrogen, Europe) of the EdU labeling kit. The Click-iT EdU reaction buffer (Invitrogen, Europe) was mixed with Alexa Fluor azide ('click' reaction) dye to detect cells that were replicating DNA. Staining was performed by freeze cracking worms after fixation with 2% formaldehyde.

## *C. elegans* tiling array analysis

Parental animals were raised at 15°C and shifted to 25°C for egg laying. Offspring was dissected after the L4-adult transition. Fifty gonads per analyzed genotype in triplicates were dissected in M9 containing levamisole. RNA was isolated using the PicoPure RNA Isolation Kit (Applied Biosystems) according to the manufacturer's recommendations. Total RNA (200 ng for the Notch ON/OFF experiments or 100 ng for the *mes*/wild-type experiments) was amplified using the Affymetrix GeneChip WT Amplified Double Stranded cDNA Synthesis Kit according to the manufacturer's instructions. The Affymetrix GeneChip WT Double-Stranded cDNA Terminal Labeling Kit was used for the fragmentation and labeling of 7.5 μg cDNA. The labeled material was loaded onto Affymetrix GeneChip C. elegans Tiling 1.0R Arrays and hybridized at 65°C for 16 hrs. The arrays were washed in an Affymetrix Fluidics station with the protocol FS450_0001 and scanned in an Affymetrix GeneChip Scanner 3000 with autoloader using Affymetrix GCC Scan Control v. 3.0.0.1214 software. All tiling arrays were processed in R (32,33) using Bioconductor (34) and the packages tilingArray and preprocessCore. The arrays were RMA background corrected and log2 transformed on the oligo level using the following command:

expr <- log2 (rma.background.correct (exprs (readCel2eSet (filenames,rotated = TRUE)))).

We mapped the oligos from the tiling array (bpmap file from www.affymetrix.com) to the *C. elegans* genome assembly ce6 (www.genome.ucsc.edu) using bowtie (*Langmead et al., 2009*), allowing no error and unique mapping position. Expressions for individual transcripts were calculated by

intersecting the genomic positions of the oligos with transcript annotation (WormBase WS190) and averaging the intensity of the respective oligos. For the *mes-4*/wt experiment, no quantile normalization was performed as the distributions for the *mes-4* mutant and the wt differed substantially. In the case of the Notch ON/OFF dataset quantile normalization was performed on the level of transcripts. The p-values were calculated in R with the hypergeometric distribution function 'phyper'.

## Validation of array analysis by RT-qPCR

RNA was isolated as described above. cDNA was synthesized with oligo dT primer using the ImProm II Reverse Transcription System from Promega according to the manufacturer's instructions. cDNA was used for qPCR with the Absolute QPCR SYBR Green ROX Mix (AbGene) on an ABI PRISM 7700 system (Applied Biosystems). qPCR reactions were performed as described previously (*Biedermann et al., 2009*). At least one primer in each pair is specific for an exon-exon junction. Sequences of the used primers are described in detail in the *Supplementary file 3C*. Mouse RNA was added before RNA isolation and reverse transcription, allowing for normalization to *cyt-c* and thereby correcting for variations in RNA isolation and reverse transcription. Standard curves for quantification were generated from a serial dilution of input cDNA for each primer pair. The amount of target present in each replicate was derived from the standard curve; an average was calculated for each of the triplicates. To compare total mRNA levels, the qPCR results were normalized to mouse *cyt-c* and to the *C. elegans* tubulin gene *tbb-2*, and fold enrichments were calculated.

## In situ hybridization for *utx-1*

RNA in situ hybridization was performed and analyzed as previously described (*Biedermann et al., 2009*). Description of the primer pairs to generate probes from cDNA can be found in the *Supplementary file 3C*. Images were captured with a Zeiss AxioImager Z1 microscope, equipped with a Zeiss AxioCam MRc camera. Images were taken in linear mode of the Axiovision software (Zeiss) and processed with Adobe Photoshop CS4 in an identical manner.

## Chromatin immunoprecipitation and qPCR (ChIP-qPCR)

ChIP was performed as described (*Askjaer et al., 2014*). In brief, worms (strains OP591, RRID:WB_OP591, and BAT890) were washed with M9 buffer and frozen as 'worm popcorn' in liquid nitrogen prior to pulverization with a mortar and pestle. The powder was dissolved in 10 volumes of 1,1% formaldehyde in PBS + 1 mM PMSF and incubated 10 mins with gentle rocking at room temperature with subsequent quenching for 5 mins at room temperature by adding 2,5 molar glycine (final concentration 125 mM). After centrifugation with 4.000 g at 4°C the pellet was washed with ice-cold PBS+1 mM PMSF and resuspended in 50 mM FA buffer (50 mM HEPES/KOH pH 7,5; 1 mM EDTA; 1% Triton-X 100; 0,1% sodium deoxycholate; 150 mM NaCl) + 1% sarkosyl + protease-inhibitors. Samples were sonicated twice using the Biorupter (15 times 30 s ON, 30 s OFF) on high settings at 4°C followed by spinning at 13.000 g, 15 min, 4°C. The supernatant corresponding to 4 mg of protein measured by Bradford assay was used for ChIPs. Samples were incubated with 50 µl of FLAG or HA antibodies coupled to µMACS microbeads (Milteny) (all blocked with 5% milk in FA-buffer). After incubating 1 hr at 4°C, the beads where washed in µMACS matrix columns in a magnetic rack with FA buffer containing 1 M and 0.5 M NaCl and subsequent wash with TE and TEL buffer (0,25 M LiCl; 1% Sodium deoxycholate; 1 mM EDTA; 10 mM Tris pH 8,0). Bound material was eluted with 65°C pre-warmed 125 µl ChIP elution buffer (1% SDS, 250 mM NaCl, 10 mM Tris pH 8.0, 1 mM EDTA) and fixation was reversed using 2 µl of 10 mg/ml Proteinase K at at 50°C for 1 hr followed by 65°C incubation overnight. DNA was purified using the Quiaquick PCR purification kit in a final volume of 40 µl and 1 µl was used for qPCR. Negative controls were used to assess the quality of the ChIP and fold enrichment of the target genes: lysates (N2 worms) which do not express the recombinant target protein with specific antibody (anti-FLAG coupled to µMACS beads) and lysates of worms expressing the recombinant target protein with unspecific antibody (anti-HA coupled to µMACS beads) controls, respectively. Primer for qPCRs was designed using Primer3Plus (*Untergasser et al., 2007*) with the following settings: amplified region min. 100 bp – max. 200 bp; GC content: 50–60%; min. primer length: 18 nt – max. length 24 nt; melting temperature: min. 58°C – max. 63°C; max.; 3' self complementary allowance set to 1; max. allowed length of a mononucleotide repeat (max. poly-x): 3. Sequences of the used primers are listed in in the *Supplementary file 3C*. The

qPCRs were run on CFX96 Touch Real-Time PCR Detection System from BioRad using the Maxima SYBR Green/ROX qPCR Master Mix (2X). The data analysis was performed by calculating the $\Delta\Delta$Ct-values. Differences were assessed using Student's $t$-test.

## DAPI staining and counting of germ cells

Worms were transferred to a slide and fixed by adding 10–20 µl 95% ethanol and letting evaporate the ethanol. The ethanol fixation was repeated 2 more times before adding the DAPI solution in microscopy mounting media (vectashield from Vector or similar). The samples were sealed with a coverslip and nail polish before microscopy. Fluorescent micrographs were recorded with Zeiss Axi-oImager Z1 microscope and the SensicamII camera (PCO) and the micromanager software was used to capture Z-stack images with 0.5 µm slice steps. Images subject to direct comparison were taken at identical exposure times. Counting of germ cells within the range from the DTC to the turn of gonadal arms of *glp-1(ar202); hsp::che-1; gcy-5::gfp* animals treated with either mock or *lin-53* RNAi was performed using the Z-stacks. Micromanager was used to control the Z-stack levels and the ImageJ plugin for cell counting for scoring the number of germ cells. Germ cell counts in gonads of Notch ON: *gld-2(q497) gld-1(q485); glp-1(ar202)* and Notch OFF: *gld-2(a497) gld-1(q485); glp-1 (e2144)* (*Figure 1—figure supplement 2*) and germline tumor phenotype in *glp-1(ar202)* and *glp-1 (ar202); rrf-1(pk1417)* were scored after dissection, formaldehyde fixation and DAPI staining. For Notch ON and Notch OFF mutants, the central plane of the gonads was imaged and germ cells in the entire dissected gonad were counted using the CellCounter plugin with ImageJ. For each of the two strains, germ cells in the entire gonad of 15 dissected gonads were counted.

## Immunostaining and antibodies

Antibody stainings of intact worms were performed using a freeze-crack procedure as described (*Duerr, 2006*). In brief, after washing, worms were resuspended in 0.025% glutaraldehyde, and frozen between two frost-resistant glass slides on dry ice. Separating the glass slides while frozen creates additional cracks in the cuticle. Acetone/methanol or 4% paraformaldehyde in 0.1 M phosphate buffer for 1 hr on ice fixation was used. Worms were washed off the slides in PBS, blocked with 0.2% gelatin + 0.25% Triton in PBS, and stained. Primary antibodies were diluted in PBS with 0.1% gelatin and 0.25% Triton and fixed worms were incubated 4 hrs - overnight at 4°C. After PBS washes secondary antibody was applied for 3 hrs. After PBS washes worms were mounted with DAPI-containing mounting medium (Dianova, #CR-38448) on glass slides. The primary antibodies used were anti-H3K27me3 (1:400; gift from Dr. Hiroshi Kimura); anti-HA (1:100, Roche #12CA5; acetone fixation); anti-H3Ser10ph (1:400, Abcam, #ab14955; acetone fixation). Secondary antibodies were Alexa Fluor dyes applied at 1:1000 dilution.

Stainings for H3K27me3 on dissected gonads were performed using anti-H3K27me3 from Millipore (catalogue number 07–449, Lot 1959680; courtesy of Jan Padeken, Gasser laboratory) on dissected gonads. The adult animals were dissected in M9 containing levamisole, fixed with 2% paraformaldehyde in PBS on poly-lysine coated slides, snap-frozen on dry ice, freeze-cracked, incubated for 5 mins in ice-cold DMF at −20°, washed for 5 mins in PBS 0.1% Tween-20 at room temperature, blocked for 20 mins in PBS 0.1% Tween-20 + 5% BSA and incubated with the primary antibody over night at 4°C. Secondary antibodies (Alexa 488, goat anti rabbit, 1:500) were applied for 2 hrs at room temperature. Slides were then washed three times for 5 mins in PBS 0.1% Tween-20 at room temperature and mounted with Vectashield mounting medium containing DAPI.

## Transgenic animals and reporter GFP quantifications

The transcriptional reporter gene *putx-1::gfp-h2b (rrrSi185)* was constructed from the 1302 bp putative promoter region of the gene *utx-1* (*human UTX (Ubiquitously transcribed TPR on X) homolog - 1*) fused to sequences encoding for GFP-H2B and the ubiquitously expressed *tbb-2* 3'UTR using the Gateway Reporter Cloning System (*Merritt et al., 2008*). The reporter gene *putx-1::gfp-h2b (rrrSi185* and *rrrSi281)* was constructed with the following primers:

putx-attB4: GGGGACAACTTTGTATAGAAAAGTTGGGATTTTATCTTCATCGGACCTG
putx-attB1 : GGGGACTGCTTTTTTGTACAAACTTGTGGCGGTGTGAGAAGCGATAC
The full-length functional transgene *putx-1::flag-gfp-utx-1::utx-1 3'UTR (rrrSi189)* was constructed with the following primers: utx-1+3UTR+attB2 L

ggggacagctttcttgtacaaagtggACGACGAATCAGAACCTCTGCCGGAGGAGCGTCATgtaag
utx-1+3UTR+attB3 R ggggacaactttgtataataaagttgaatgcggatactgccttctc

The functional UTX-1 transgene *putx-1::flag-gfp-tev::utx-1::utx-1 3'utr(rrrSi189)* contains the same promoter as the transcriptional reporter, the full-length *utx-1* genomic sequence as well as the endogenous 3'UTR, and was equipped with N-terminal GFP, FLAG, and TEV tags. Transgenic animals were produced as single-copy integrants using the MosSCI direct insertion protocol (*Frøkjaer-Jensen et al., 2008*). The *rrrSi189* transgene is functional, as it rescues the *utx-1* mutant (*ok3553* allele). For GFP quantification, gonads were dissected from live animals in M9 buffer containing levamisole and mounted to glass microscopy slides (*Frøjkaer-Jensen et al., 2008*). Fluorescent micrographs were recorded with Zeiss AxioImager Z1 microscope and a Zeiss Axioncam MRm REV 2 CCD camera was used to capture images. Fluorescence intensities were quantified using ImageJ. GFP intensities were normalized to the picture background and corrected with the average auto-fluorescences measured in wild type (N2) gonads at the corresponding temperatures. Images subject to direct comparison were taken at identical exposure times and were processed with Adobe Photoshop CS4 in an identical manner. The numbers of analyzed gonads were as follows: n = 44 for wild-type reporter; n = 36 for *glp-1(gf* $^{ts}$*)*; n = 55 for wild-type reporter on control RNAi; n = 48 for *mes-2(RNAi)*; n = 15 for *mes-3(RNAi)*; n = 29 for *mes-6(RNAi)*, and n = 20 for LAG-1 binding sites deleted reporter.

## Genetic interaction of *glp-1* and *mes-2*

Alleles used were *glp-1(ar202)* and *mes-2(bn11)*. Worms were grown at the semi-permissive temperature of 20°C and gonads were dissected and DAPI-stained shortly after the L4-young adult transition.

The experiment was performed twice. In a first round, a low number of gonads were examined to identify whether the double mutant had a phenotype and to define phenotypic categories to score. Based on the observation of a clear and penetrant phenotype, gonads were scored in a second round according to the categories defined in the first round.

## Acknowledgements

We thank Sergej Herzog, Alina El-Khalili, Mei He, Sandra Muehlhaeusser and Iskra Katic for technical assistance, and the following FMI technology platforms: functional genomics, bioinformatics, advanced imaging and microscopy. We also thank the CGC, supported by the NIH, Tim Schedl and Dave Hansen for providing strains. We thank Susan Gasser, James Priess, Dirk Schübeler, Gunter Merdes and members of the Tursun and Ciosk groups for discussion and comments on the manuscript. This work was partly sponsored by the SBFI grant Nr. C15.0038 to RC. The Friedrich Miescher Institute for Biomedical Research is sponsored by the Novartis Research Foundation. BT receives funding from ERC-StG-2014-637530 and ERC CIG PCIG12-GA-2012-333922 and is supported by the Max Delbrueck Center for Molecular Medicine in the Helmholtz Association.

## Additional information

### Funding

| Funder | Grant reference number | Author |
| --- | --- | --- |
| European Research Council | ERC-2014-STG #637530 - REPROWORM | Baris Tursun |
| European Research Council | FP7-PEOPLE-2012-CIG #333922- REPROL53U48 | Baris Tursun |
| European Cooperation in Science and Technology | SBFI C15.0038 | Rafal Ciosk |

The funders had no role in study design, data collection and interpretation, or the decision to submit the work for publication.

## Author contributions
SS, IA-K, Acquisition of data, Analysis and interpretation of data, Drafting or revising the article; BH, MH, SG, Acquisition of data, Analysis and interpretation of data; BT, RC, Conception and design, Acquisition of data, Analysis and interpretation of data, Drafting or revising the article, Contributed unpublished essential data or reagents

## Author ORCIDs
Baris Tursun, http://orcid.org/0000-0001-7293-8629
Rafal Ciosk, http://orcid.org/0000-0003-2234-6216

## Additional files

### Supplementary files
• Supplementary file 1. Enhancement or suppression of tumorous phenotypes.

• Supplementary file 2. In situ hybridization patterns of Notch-activated genes.

• Supplementary file 3. Information on used *C. elegans* strains, RNAi clones and primers. (A) Information on *C. elegans* strains used in the study. (B) Information on RNAi clones used in this study. (C) Information on primer design and sequences.

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
