## [Decision Letter]

Thank you for submitting your article "Notch signaling antagonizes PRC2-mediated silencing to promote reprograming of germ cells into neurons" for consideration by *eLife*. Your article has been reviewed by three peer reviewers, and the evaluation has been overseen by a Julie Ahringer as the Reviewing Editor and Kevin Struhl as the Senior Editor. The reviewers have opted to remain anonymous.

The reviewers have discussed the reviews with one another and the Reviewing Editor has drafted this decision to help you prepare a revised submission.

Summary:

In this study the authors investigate the reprogramming of germ cells into neurons in the adult *C. elegans* germ line upon removal of certain chromatin modifiers, an in vivo reprogramming process they refer to as GeCo. Here they show a requirement for Notch signalling in this process, and suggest that this role of Notch is independent of its role in promoting proliferation. They also identify targets of Notch, one of which is an H3K27 demethylase, UTX-1, suggesting a mechanism by which Notch-directed transcription may counteract PRC2-dependent repression.

Essential revisions:

1) The claim that the antagonism between GLP-1 Notch signaling and PRC2 that you study here is functionally relevant in the germline, or conceptually, in reprogramming, is not substantiated. In particular, the X-linked GLP-1 transcriptional targets you studied (e.g., *utx-1*) do not appear to be GLP-1 transcriptional targets in normal germline development, although they may become GLP-1 targets in the absence of PRC2. The data seem consistent with GLP-1 signaling acting within an abnormal permissive context generated by loss of PRC2. The paper should be reframed to address this point or the claim should be justified.

2) The authors assume that genetic perturbations employed are germline autonomous, but do not experimentally demonstrate that this is the case. The *mes* genes function in the soma (Ross and Zarkower, 2003). It has also been shown that loss of gene activity in the somatic gonad can result in enhancement of the *glp-1(ar202*) tumorous germline phenotype (McGovern et al., 2009). Therefore, the authors should show that the *mes* gene mutant enhancement of *glp-1(ar202)* is germ cell autonomous, which is required based on the mechanism proposed. Similarly, the authors use RNAi of Notch upregulated genes to substantiate the argument that the genes are relevant for germ cell proliferation. While data from others has shown that some of the identified genes function in the germline (for example, Kershner et al., 2014; Doyle et al., 2000) it is also well known that loss of function of genes that function only in the soma can result in reduced proliferation (for example, Dalfo et al., 2012). Note that for *utx-1*, its function in reprogramming is interesting whether it acts in the germline or the soma, although models for its action would be different.

3) The authors conclude that the GeCo phenotype requires Notch signaling, independent of proliferation. To be confident in the conclusion, the following issues should be addressed.

A) As *lag-1* is the transcriptional effector of Notch signaling in *C. elegans*, Figure 1 does not give confidence that the *lag-1* RNAi is working as the *lag-1* RNAi should cause the *glp-1* gain of function germ cells to enter meiosis, resulting in fewer germ cells and a different germline morphology compared to the mock treated (Berry et al., 1997).

B) From the images in Figure 1—figure supplement 3, it appears that the HU treatment may have not been effective. HU should inhibit mitosis in the germ line, leading to fewer germ cell nuclei, and it should induce cell cycle arrest, leading enlarged germ nuclei. Although the HU image in Figure 1—figure supplement 3 shows a lack of Edu positive nuclei, there is no evidence of enlarged germ cell nuclei or of reduced germ cell number, which is expected for HU treatment (see Figure 4 in Gartner et al., 2004). The control for HU treatment is also not adequately explained.

C) What developmental or cell cycle state are the germ cells with more or less Notch activity? This is important since Notch regulates the proliferative state of germ cells. Cells that are pre-meiotic may not be able to convert readily compared to proliferative germ cells. Note that this is independent of cell numbers, but relies on markers and morphology of the germ line. Conversely, in the HU block experiment, do the germ cells enter the meiosis pathway?

D) Although they should be related, the assessment of total germ cell number is not a measure of mitotic germ cell number, which is the key measure of proliferation. To determine whether the increase in conversion is a consequence of an increase in number of mitotic germ cells, the authors should quantify the number of mitotic germ cells (e.g., using phospho histone H3 staining) rather than total germ cell number.

4) It would improve the paper to show a stain of H3K27me3 in the different cellular environments (Notch single and triple mutants) and to compare the location of H3K27me3 with locations that can respond to heterologous regulators.

5) H3K27me3 might be predicted to silence transgenes introduced into worms, such as those involved in reprogramming. The authors should check the level of CHE-1 expression in the different cellular environments to make sure that the changes they track reflect the response of the tissue and not the expression of the inducer.

6) The authors should improve the citations for previous *C. elegans* studies on Notch, PRC2, and reprogramming in *C. elegans* (e.g. from the Mango, Rothman, Krause, Priess labs such as Djabrayan et al., Genes & Development, 2012, Yuzyuk et al., Dev Cell, 2009).

[Editors' note: further revisions were requested prior to acceptance, as described below.]

Thank you for resubmitting your work entitled "Increasing Notch signaling antagonizes PRC2-mediated silencing to promote reprograming of germ cells into neurons" for further consideration at *eLife*. Your revised article has been favorably evaluated by Kevin Struhl (Senior editor), a Reviewing editor, and three reviewers.

The manuscript has been improved but there are some remaining issues that need to be addressed before acceptance, as outlined below. Two are particularly important:

1) Your new experiments do not convincingly show that the interaction between Notch and PRC2 is germ line autonomous, so it remains uncertain where PRC2 and *utx-1* are acting in *glp-1(ar202)* proliferation and conversion into neurons. Gene activity in the soma can modify *glp-1(ar202)* overproliferation in the germline.

2) Second, the conclusion that UTX-1 is a germ line target of Notch is not convincing.

These and the other points raised in the attached reviews all need to be addressed, either by experiments or by changing your text. Please note that we can offer no further opportunities for additional work or modification thus the binding decision must be made on your next revised submission.

Additionally, points in your paper overstated or incorrect as described in the reviews. For example, Vandamme et al. 2012 did not show endogenous UTX-1 expression in the germ line, contrary to what is stated in your paper. In other places there is insufficient information provided to allow readers outside of the immediate field to understand what is already known and what then is an expected result.

*Reviewer #1:*

The investigation of the cell cycle vs Notch effects is carefully done and convincing.

The question of cell autonomy is not convincing (Figure 2—figure supplement 3 and Figure 5). In particular:

The authors use *rrf-1* to address whether genes function within the germ line. The logic is that a germline autonomous regulator should not be perturbed by the addition of an *rrf-1* mutation. This is a weak line of reasoning as *rrf-1* is required in some somatic tissues but not others: See Kumsta and Hansen 2012.

In addition, mes-2 looks affected by *rrf-1* mutations. So is this actually a germ-line role of PRC2? Or could it be somatic? And why do *mes-3* and *mes-6* look different from *mes-2*?

*rrf-1* appears to modify the tumorous phenotype on its own. Why do *rrf-1* mutants suppress *glp-1* Tumors?

Finally, it is unclear how the authors are calling the cut off for germline autonomous vs. somatic effects. Not only for *mes-2* but for all genes.

A simpler way to address this question would be to express a rescuing construct in the germ line alone. E.g. a mini Mos construct. This is most critical for *utx-1*, in my opinion.

The conclusion that UTX-1 acts in the germ line is also not convincing for three reasons.

First, Seelk et al. use the promoter of *utx-1* with a histone GFP readout (Figure 5). While this shows changes in *glp* and *mes* mutants, it is convincing with regard to UTX activity because the germ line relies heavily on post-transcriptional regulation. To ensure that the protein reflects RNA levels, the authors need to look at the endogenous protein or a single copy, rescuing reporter that has the whole protein and regulatory sequences.

Second, Vandamme 2012 argued that UTX-1 activity in the soma influenced the germ line, perhaps through the distal tip cell. The distal tip cell signals through the Notch pathway, so this is a reasonable idea.

Third, Vandamme 2012 did not show endogenous UTX-1 expression in the germ line, as stated by Seelk et al.

The key question is whether UTX and H3K27me3 function in the germ line and downstream of Notch to control reprogramming via target gene regulation. It may be that some events occur in the soma, so I don't think the authors need to test everything for cell autonomy, but *utx-1* would be useful, particularly given previous publications suggesting at distal tip cell focus.

The H3K27me3 stains appear to have been done without controls, so it is impossible to judge if there are any changes in different strain backgrounds. (If there were, it would be more convincing that *utx* was acting in the germ line.) A well-controlled experiment would include an on-slide control (e.g. a marked N2) and a germline control (e.g. histone H3). The ratio of H3K27me3:H3 would give an indication of whether levels are changing.

References to be added:

Introduction, second paragraph: add Horner et al., 1998 as one of the first examples of reprogramming in an intact organism and of timing of competence.

Introduction, second or third paragraph: add Yuzyuk 2009 for PRC2 -dependent termination of plasticity and promotion of cell fate stabilization.

Introduction, third paragraph: add Maures 2011, Jin 2011, maybe Vandamme 2012 for identification of utx-1 as a regulator of H3K27me3.

*Reviewer #2:*

A number of changes need to be made to the manuscript so that the results and possible conclusion are more transparent.

1) Paragraph with heading “Both GLP-1^Notch^ and PRC2 regulate expression of the H3K27 demethylase UTX-1”:

A) The authors should first explain what is the expected expression pattern for a GLP-1 transcriptional target, based on the known and experimentally validated direct transcriptional targets of GLP-1 Notch in the gonad, *lst-1* and *sygl-1*, described in Kershner et al. 2015. The failure to do so makes the paper hard to understand for readers outside the immediate *C. elegans* field.

B) The expected pattern based on the work of Kershner et al., 2015, is that GLP-1 dependent transcripts are limited to the distal most 5 to 10 cell diameters, consistent with ligand expression limited to the distal tip cell. The authors interpretation of the *putx-1::gfp* (Figure 1) and *putx-1::FLAG-GFP::utx-1* (Figure 5—figure supplement 1) is that there is weak expression in the distal-most region of the germline where GLP-1 targets are expected to be expressed. However, this reviewer cannot see any *putx-1* GFP signal above background, and this view is supported by the absence of mRNA in the distal-most germline from the in situ hybridization experiments in Figure 5—figure supplement 2). Therefore, the sentence "Consistent with the expression patterns of the endogenous *utx-1* (Vandamme et al. 2012) and of a GFP used, functional *utx-1* transgene (Figure 5—figure supplement 1), the *utx-1* reporter was weakly expressed in the distal-most, proliferative part of the germline (Figure 5)." should be changed to indicate that there is "little or no expression in the distal-most, proliferative part of the germline".

C) The sentence "However, a similar expression pattern has been reported for another reported GLP-1Notch target gene, *lip-1* (Hajnal and Berset 2002; Lee et al. 2006)." should be removed. Hajnal and Berset 2002 showed that *lip-1* is a *lin-12* transcriptional target in vulval development, with no information on *glp-1* or the germline. Lee et al. 2006 did not perform the *glp-1* dependence experiment comparing *gld-1 gld-2* with *gld-1 gld-2; glp-1*, which the authors in this manuscript performed, and thus it is uncertain if *lip-1* is a germline target of GLP-1 in distal-most germ cells.

D) In the first two paragraphs of the subsection “UTX-1 is required for GLP-1^Notch^-mediated GeCo enhancement”. There is no evidence that GLP-1 signaling occurs in the medial/proximal gonad, as GLP-1 protein is not found in this region (Crittenden et al., 1994). Therefore, the expression of the 33 genes indicated is either an indirect consequence of GLP-1 signaling (another transcription factor that is transcribed in the distal-most germline stem cells in response to GLP-1 signaling which then acts to induce transcription much later in development, in pachytene cells) or is not regulated by GLP-1 signaling at all. The sentence needs to be changed to indicate that the observed expression is *not* consistent with direct GLP-1 transcriptional regulation.

2) In the subsection “Both GLP-1^Notch^ and PRC2 regulate expression of the H3K27 demethylase UTX-1”. "Deleting those sites drastically reduced reporter expression in the medial and proximal portions of the gonad (Figure 5)." Figure 5 indicates that the decrease is at most 25%. The sentence should be "Deleting those sites reduced reporter expression by approximately 25% in the medial and proximal portions of the gonad (Figure 5)."

3) The last paragraph of the subsection “Both GLP-1^Notch^ and PRC2 regulate expression of the H3K27 demethylase UTX-1”.

"Overall, our observations suggest that PRC2 and GLP-1^Notch^ signaling have antagonistic effects on *utx-1* transcription. However, the endogenous levels of GLP-1^Notch^ signaling are apparently insufficient to overcome PRC2-mediated repression in the distal part of the gonad" does not emphasize the findings in the manuscript and should be rewritten in the following way:

"Overall, our observations suggest that PRC2 and GLP-1^Notch^ signaling have antagonistic effects on *utx-1* transcription that can be observed in a PRC2 mutant background. However, in wild type the endogenous levels of GLP-1^Notch^ signaling are apparently insufficient to overcome PRC2-mediated repression in the distal part of the gonad".

4) Discussion. While the sentence "This suggests that the identified set of genes may contain additional physiological GLP-1Notch targets." may be true, of the genes analyzed in this manuscript, none have been shown to be convincing GLP-1^Notch^ targets in wild type animals.

The Discussion should instead focus on their novel findings – that PRC2 is apparently acting antagonistically to prevent certain GLP-1^Notch^ targets from being expressed in the germline, which makes sense as it is likely important to block the expression of some somatic GLP-1 targets from being expressed in germ cells, and that these include *utx-1* which promotes reprogramming.

*Reviewer #3:*

I think in general the study stands up as providing evidence that there is some antagonism between Notch activation and PRC2-mediated repression of genes, and that enhancement of Notch activity in germ cells yields similar transcriptional consequences to loss of PRC2 repression in germ cells – enhanced X-linked gene expression. Similarly, these conditions also both make it easier to drive germ cells towards somatic differentiation upon ectopic activation of somatic fate drivers, such as *che-1*.

The question still stands whether this is informative for either normal germ cell development or for understanding the balance of activation and repression controlling early embryo development or programming/reprogramming.

I think it's clear that the processes studied are artificial in terms of germ cell development – but not entirely so. I think the authors do show that the balance of Glp-1 signaling and PRC2 repression are essential for normal germ cell development. I also think they've shown they can uncouple some of the Glp-1-dependent effects from proliferation phenotypes. This probably underlies the differences in absolute phenotypes between Glp-1 gof and PRC2 mutants (which are presumably maternally rescued mutants): proliferation defects in the former compared to little visible defects in the maternally rescued germ cells. It's formally interesting that raising the level of Notch activity has the same effect on transcriptional regulation as lowering PRC2 activity in these cells. It's interesting that this seems to be separate from any need for proliferation to be dysregulated.

I think the authors have adequately addressed the technical concerns about this conclusion.

There still is a bit of what I view as semantic issues with how some things are phrased that can be over-interpreted by the reader, or may indeed still be overstatements by the authors:

For example, the subsection “GLP-1^Notch^ and PRC2 have antagonistic functions in the germline” states "that, in the *C. elegans* gonad, specific PRC2-repressed genes are activated via GLP-1^Notch^ signaling." This certainly implies that *glp-1* activation of genes like *utx-1* is part of normal germ cell function, which is unlikely to be true.

Lastly, the description of the *mes* mutants used needs to be clarified. I am assuming that when the authors are referring to *mes-2(bn11)* mutants, etc. they are actually referring to the F1 maternally rescued homozygote offspring (e.g., M Z ). It is important to clarify that these animals and their germ cells are not completely devoid of a history of PRC2 activity, which is why they have germ cells to count and look at in the first place. Absent maternal MES protein, there wouldn't be but a few dying germ cells in the adult gonads to look at.

---

## [Author Response]

*Essential revisions:*

*1) The claim that the antagonism between GLP-1 Notch signaling and PRC2 that you study here is functionally relevant in the germline, or conceptually, in reprogramming, is not substantiated. In particular, the X-linked GLP-1 transcriptional targets you studied (e.g., utx-1) do not appear to be GLP-1 transcriptional targets in normal germline development, although they may become GLP-1 targets in the absence of PRC2. The data seem consistent with GLP-1 signaling acting within an abnormal permissive context generated by loss of PRC2. The paper should be reframed to address this point or the claim should be justified.*

We would like to point out that, despite using compound mutants to uncover putative GLP-1 targets, these strains contain wild-type PRC2. Thus, *utx-1*, along with at least some other identified genes, such as the confirmed targets *sygl-1* and *lst-1* (Kershner et al. 2014), is regulated in the context of normal PRC2. In agreement with GLP-1– dependent regulation of *utx-1* in normal development, *utx-1* is bound by LAG-1 and the deletions of LAG-1 binding sites in the *utx-1* promoter, expressed in the wild-type background, impair its expression.

It is true, however, that whether normal levels of Notch signaling are sufficient to antagonize PRC2 repression in normal development remains uncertain. We made this point clear at the end of the Results section, stating: “However the endogenous levels of GLP-1/Notch signaling are apparently insufficient to overcome PRC2-mediated repression (of *utx-1*) in the distal part of the gonad.”

*2) The authors assume that genetic perturbations employed are germline autonomous, but do not experimentally demonstrate that this is the case. The mes genes function in the soma (Ross and Zarkower, 2003). It has also been shown that loss of gene activity in the somatic gonad can result in enhancement of the glp-1(ar202) tumorous germline phenotype (McGovern et al., 2009). Therefore, the authors should show that the mes gene mutant enhancement of glp-1(ar202) is germ cell autonomous, which is required based on the mechanism proposed. Similarly, the authors use RNAi of Notch upregulated genes to substantiate the argument that the genes are relevant for germ cell proliferation. While data from others has shown that some of the identified genes function in the germline (for example, Kershner et al., 2014; Doyle et al., 2000) it is also well known that loss of function of genes that function only in the soma can result in reduced proliferation (for example, Dalfo et al., 2012). Note that for utx-1, its function in reprogramming is interesting whether it acts in the germline or the soma, although models for its action would be different.*

To address this valid concern, we used the *glp-1(ar202); rrf-1(pk1417)* strain to re-screen *mes* genes and several enhancers/suppressors for germ cell autonomy. The *rrf-1* mutant strains have impaired somatic RNAi response, while the germline RNAi response is still robust (Sijen et al., 2001).

While some enhancers, such as *mbk-1* and B0416.5 seem to work from the somatic tissues (i.e. enhance the tumorous phenotype *in glp-1(ar202)* but not in *glp-1(ar202); rrf-1(pk1417)* mutants), other enhancers and suppressors, such as *puf-9*, are germ cell autonomous (see **F**Figure 2—figure supplement 3). Importantly, while *mes-2* and *mes-3* RNAi knockdowns were not very efficient (this has been observed in the lab before) and did not enhance the tumorous phenotype in either strain, RNAi-mediated depletion of MES-6 did enhance the phenotype in both *glp-1(ar202)* and *glp-1(ar202); rrf-1(pk1417)* strains. Thus, while there may be a somatic role for some of the enhancers or suppressors, the genetic interaction between GLP-1 and PRC2 is autonomous to the germline.

*3) The authors conclude that the GeCo phenotype requires Notch signaling, independent of proliferation. To be confident in the conclusion, the following issues should be addressed.*

*A) As lag-1 is the transcriptional effector of Notch signaling in C. elegans, Figure 1 does not give confidence that the lag-1 RNAi is working as the lag-1 RNAi should cause the glp-1 gain of function germ cells to enter meiosis, resulting in fewer germ cells and a different germline morphology compared to the mock treated (Berry et al., 1997).*

As pointed out, efficient depletion of LAG-1 will result in precocious differentiation of germ cells or suppress *glp-1* gf tumors. Indeed, *lag-1* RNAi was our control in the screen for *glp-1* gf tumor suppressors. However, a weaker *lag-1* RNAi was precisely the strength of our experiments, separating the effect on GeCo from the effect on proliferation. Our protocol involved *lag-1* RNAi only after hatching (P0 RNAi). By this, we bypassed the L1 lethality as well as the *glp-1(lf)-*like phenotype (loss of mitotic germ cells). Additionally, our use of double RNAi combinations (either with control vector or *lin-53* RNAi) in the *lag-1* assay effectively diluted the RNAi 1:1. Thus, we did not observe a fully penetrant *lag-1* phenotype for germ cell proliferation. Avoiding the *glp-1(lf)-*like phenotype was important, since we showed previously that the *glp-1(lf)* mutant background results in the loss of GeCo (Tursun et al. 2011). We apologize that we failed to explain this sufficiently. We state now explicitly in the Results section that “We exposed animals only after hatching to *lag-1* RNAi in order to avoid sterility[…]”

*B) From the images in Figure 1—figure supplement 3, it appears that the HU treatment may have not been effective. HU should inhibit mitosis in the germ line, leading to fewer germ cell nuclei, and it should induce cell cycle arrest, leading enlarged germ nuclei. Although the HU image in Figure 1—figure supplement 3 shows a lack of Edu positive nuclei, there is no evidence of enlarged germ cell nuclei or of reduced germ cell number, which is expected for HU treatment (see Figure 4 in Gartner et al., 2004). The control for HU treatment is also not adequately explained.*

The protocol by Gartner, 2004 uses a 12 h incubation and subsequent EdU staining, which results in both, loss of EdU staining and the enlargement of germ cells. We have performed a new experiment shown in the Figure 1—figure supplement 3using 12 h incubation with HU. We can see the enlargement of nuclei based on DAPI as described by Gartner, 2004 and Fox et al., 2011. We also performed H3Ser10ph (pH3) antibody staining and the loss of pH3 staining in the HU treated *glp-1(gf)* germline suggests that the germ cells undergo cell cycle arrest, indicating that HU treatment is, also in our hands, effective.

Importantly, Fox et al. 2011 have described that five hours of HU treatment are sufficient to abolish EdU staining, while germ cells do not yet enter a premeiotic stage. In a previous study using the exact same conditions, we could see the depletion of EdU incorporation but no obvious nuclei enlargement after 5 h (Patel et al. 2012). Therefore, we decided to use a 5 h exposure for imaging the cell cycle arrest in our original Figure 1—figure supplement 3 panel B. While we cannot see obvious changes in germ cell nuclei size in the *glp-1(gf)* background after 5 h, we are confident that we have effectively induced cell cycle arrest by that time. We have further clarified our experimental procedure in Materials and methods / Cell cycle arrest by HU treatment and EdU staining.

*C) What developmental or cell cycle state are the germ cells with more or less Notch activity? This is important since Notch regulates the proliferative state of germ cells. Cells that are pre-meiotic may not be able to convert readily compared to proliferative germ cells. Note that this is independent of cell numbers, but relies on markers and morphology of the germ line. Conversely, in the HU block experiment, do the germ cells enter the meiosis pathway?*

We agree that GeCo diminishes when germ cells prematurely enter meiosis as we have shown before using the *glp-1(lf)* background (Tursun et al., 2011). The *glp-1(lf)* background is known to cause a premature entry into meiosis while *glp-1(gf)* causes a tumorous germ line due to increased proliferation and expansion of the meiotic zone. The tumorous germlines of *gld-1 gld-2* and of *gld-1 gld-2; glp-1(lf)* mutants have been characterized previously by Kadyk et al., 1998 and Hansen et al., 2004 using pH3, REC- 8, and HIM-3 antibodies. While wild-type germlines or *glp-1(lf)* germlines show 100% entry into meiosis, in the double or triple mutants, only 10 – 20% of cells express early meiotic markers. This argues against the possibility that the loss of GeCo in *gld-1 gld-2; glp-1(lf)* triple mutant germlines is caused by increased meiotic entry because these cells actually barely enter into meiosis. We added this information to the revised manuscript.

*D) Although they should be related, the assessment of total germ cell number is not a measure of mitotic germ cell number, which is the key measure of proliferation. To determine whether the increase in conversion is a consequence of an increase in number of mitotic germ cells, the authors should quantify the number of mitotic germ cells (e.g., using phospho histone H3 staining) rather than total germ cell number.*

In order to disentangle whether ‘increased proliferation’ versus ‘Notch signaling per se’ is required for GeCo enhancement, we made use of the triple mutant background *gld-1 gld-2; glp-1(lf),* in which germ cells proliferate independently from GLP-1. This strain has been extensively documented to have a tumorous germline due to over-proliferation of the germ cells (Kadyk et al., 1998; Hansen et al., 2004; Fox et al., 2011). As shown in Figure 1 and Figure 3, GeCo is almost lost in this triple mutant due to the absence of GLP-1. Hence, an increase in mitotic germ cells is not sufficient to result in the increase of conversion. Also, inhibiting the cell cycle with HU (which abolishes PH3-positive cells as shown in Figure 1—figure supplement 3using phospho histone H3 staining) did not abolish GeCo, which suggests that the GLP-1 dependent enhancement is cell cycle independent.

*4) It would improve the paper to show a stain of H3K27me3 in the different cellular environments (Notch single and triple mutants) and to compare the location of H3K27me3 with locations that can respond to heterologous regulators.*

Based on communication with the Editor, we have performed H3K27me3 staining of gonads from different *glp-1* backgrounds. We did not detect any obvious changes in the global staining, consistent with GLP-1-dependent activation of a subset of PRC2-repressed genes. We provided this data in the new Figure 2—figure supplement 4.

*5) H3K27me3 might be predicted to silence transgenes introduced into worms, such as those involved in reprogramming. The authors should check the level of CHE-1 expression in the different cellular environments to make sure that the changes they track reflect the response of the tissue and not the expression of the inducer.*

To address this valid concern, we performed HA antibody staining for the 3xHA-tagged CHE-1 protein, which is being induced after heat-shock treatment in the different genetic backgrounds. We could not detect obvious changes in the induction of CHE-1::3xHA in the germlines of the different genetic backgrounds. Hence, the different effects on GeCo does not seem to be due to changes in the CHE-1 expression levels. We provide this data in the new Figure 4—figure supplement 1.

*6) The authors should improve the citations for previous C. elegans studies on Notch, PRC2, and reprogramming in C. elegans (e.g. from the Mango, Rothman, Krause, Priess labs such as Djabrayan et al., Genes & Development, 2012, Yuzyuk et al., Dev Cell, 2009).*

We apologize for incorrect citations and have corrected them throughout the manuscript.

[Editors' note: further revisions were requested prior to acceptance, as described below.]

*The manuscript has been improved but there are some remaining issues that need to be addressed before acceptance, as outlined below. Two are particularly important:*

*1) Your new experiments do not convincingly show that the interaction between Notch and PRC2 is germ line autonomous, so it remains uncertain where PRC2 and utx-1 are acting in glp-1(ar202) proliferation and conversion into neurons. Gene activity in the soma can modify glp-1(ar202) overproliferation in the germline.*

*2) Second, the conclusion that UTX-1 is a germ line target of Notch is not convincing.*

*These and the other points raised in the attached reviews all need to be addressed, either by experiments or by changing your text. Please note that we can offer no further opportunities for additional work or modification thus the binding decision must be made on your next revised submission.*

*Additionally, points in your paper overstated or incorrect as described in the reviews. For example, Vandamme et al. 2012 did not show endogenous UTX-1 expression in the germ line, contrary to what is stated in your paper. In other places there is insufficient information provided to allow readers outside of the immediate field to understand what is already known and what then is an expected result.*

As explained in our detailed response to the Reviewers, we thoroughly re-wrote the manuscript to eliminate errors, overstatements, and to improve clarity. As for the two major points, we could answer them by improving writing and performing a new experiment.

Firstly, while using the somatic RNAi-deficient *rrf-1* mutant to probe germline autonomy has limitations, which we now explain, we can use this background to eliminate the possibility, brought up by Reviewer 1, that the examined effects originate from the somatic gonad. As published by Kumsta and Hansen 2012, RNAi is in fact defective in the somatic gonad, including in the distal tip cell (DTC), allowing this type of analysis. We now provide new data, using the *rrf-1* background, supporting the germline- autonomy of UTX-1 in reprograming. Nevertheless, as requested by Reviewers, we have softened our conclusions as explained below.

Secondly, while indeed Vandamme et al. 2012 did not show the germline expression of *utx-1* – we are sorry for this inaccuracy – this was most likely so because extrachromosomal arrays, which they used to express UTX-1, are typically silenced in the germline. In fact, one of our supplemental figures showed exactly the germline expression of single copy-integrated, FLAG and GFP tagged, rescuing UTX-1, strongly arguing that the UTX-1 protein is normally expressed in the germline. We suspect this data was misunderstood because of unclear labeling, which we have now changed, as explained below. Finally, by using the *utx-1* promoter-driven reporter, we showed that the expression of this reporter in the germline responded to the alterations in either PRC2 or GLP-1/Notch.

Our new data on UTX-1-dependent reprograming in the *rrf-1* background, combined with the germline expression of tagged, rescuing UTX-1, and the *utx-1* promoter reporter studies, corroborate now the notion that UTX-1 functions in the germline. Nevertheless, we were careful to avoid overstatements and discussed alternative scenarios, as explained below.

*Reviewer #1:*

*The investigation of the cell cycle vs Notch effects is carefully done and convincing.*

*The question of cell autonomy is not convincing (Figure 2—figure supplement 3 and Figure 5). In particular:*

*The authors use rrf-1 to address whether genes function within the germ line. The logic is that a germline autonomous regulator should not be perturbed by the addition of an rrf-1 mutation. This is a weak line of reasoning as rrf-1 is required in some somatic tissues but not others: See Kumsta and Hansen 2012.*

We thank the Reviewer for pointing this out. Indeed, while the *rrf-1* mutation was used by others to examine germline autonomy (for example Wang et al. 2012, Developmental Dynamics; Killian and Hubbard 2004, Development; Chen and Greenwald 2015, Genes Genomes Genetics; Fox et al. 2011, Development), the study by Kumsta and Hansen reported ongoing RNAi in the intestine, and residual RNAi in some hypodermal cells.

Importantly, they confirmed deficient RNAi in the somatic gonad, including the distal tip cell (DTC). Thus, while this approach is not perfect, we still consider it as our best option to address germline autonomy. Nevertheless, to avoid overstatements, we now explain the limitation of this assay.

In the paragraph “GLP-1_Notch_ activates genes mainly on X chromosomes”:

“While some of these genes may function autonomously in the germ line, others could affect the germ line indirectly from the soma. […] While depleting most candidates in the *rrf-1* background had similar effects on the germ line as in the wild type (suggesting germ line- autonomous function), in some cases the effects were abolished, suggesting that these genes function in the soma (Figure 2—figure supplement 3, [Supplementary-material SD2-data]).”

In the paragraph “UTX-1 is required for GLP-1_Notch_-mediated GeCo enhancement”: “Because UTX-1 was suggested to effect gonadal development by functioning in the somatic gonad (Vandamme et al. 2012), we re-examined GeCo efficiency upon *utx-1* RNAi in the *rrf-1 (pk1417)* background. […] Because the suppression of GeCo upon *utx-1* RNAi was observed also in the *rrf-1* background, UTX-1 does not seem enhance GeCo by functioning in the somatic gonad (Figure 4—figure supplement 1).”

In addition, mes-2 looks affected by rrf-1 mutations. So is this actually a germ-line role of PRC2? Or could it be somatic? And why do mes-3 and mes-6 look different from mes-2?

We apologize for the confusion. Apparently, our remark in the legend that, in this experiment, RNAi against *mes-2* and *-3* were ineffective, did not stand out. Since MES- 2, -3 and –6 function in a complex, and to avoid confusion, we have removed now the *mes-2* and *–3* RNAi data from the graph and re-wrote the legend accordingly (see the modified Figure 2—figure supplement 3).

*rrf-1 appears to modify the tumorous phenotype on its own. Why do rrf-1 mutants suppress glp-1 Tumors?*

This indeed appears to be the case, perhaps indicating a role for somatic endoRNAi. We presented the *glp-1(gf)* single mutant data nonetheless, to show that depleting chosen genes by RNAi had the expected effects on germline tumors (enhancement or suppression). However, the main point of this experiment was to compare the effects between different RNAi conditions in the *rrf-1* background. So, to avoid confusion, we decided to split the graphs between *glp-1(gf)* and *glp-1(gf); rrf-1* backgrounds (though we still remark in the legend to Figure 2—figure supplement 3 that the tumors were affected in the *rrf-1* background for an unknown reason):

“We observed that, for an unknown reason, the *rrf-1(pk1417); glp-1(ar202)* double mutants were less likely to produce tumors at the semi-permissive temperature of 20°C.”

*Finally, it is unclear how the authors are calling the cut off for germline autonomous vs. somatic effects. Not only for mes-2 but for all genes.*

With the caveat of the residual RNAi in the intestine, we considered an RNAi-induced effect to indicate germline autonomy when the effect was observed in both *glp-1(gf)* and *glp-1(gf); rrf-1* backgrounds. If the effect was observed in only the *glp-1(gf)* background (but not the *glp-1(gf); rrf-1* background), we considered it indicative of a somatic effect. Clearly, with increased numbers of animals, we would be able to evaluate additional borderline cases and provide statistics. However, *glp-1(gf)* animals grow very slowly (need to be propagated at 15º), so performing more replicates was, due to the revision deadline, not doable. While we may have missed minor effects, we would like to point out that the main purpose of the experiment was to demonstrate that somatic effects may play a role for *s*ome of these genes, which is the case, and that, for PRC2, somatic effects do not play a major role (which we show for *mes-6*). We hope that restructuring the figure and our explanations will satisfy the Reviewer. We now emphasize that some of the Notch activated genes may function in the soma in the paragraph “GLP-1_Notch_ activates genes mainly on X chromosomes”:

“While some of these genes may function autonomously in the germ line, others could affect the germ line indirectly from the soma. […] While depleting most candidates in the *rrf-1* background had similar effects on the germ line as in the wild type (suggesting germ line- autonomous function), in some cases the effects were abolished, suggesting that these genes function in the soma (Figure 2—figure supplement 3, [Supplementary-material SD2-data]).”

*A simpler way to address this question would be to express a rescuing construct in the germ line alone. E.g. a mini Mos construct. This is most critical for utx-1, in my opinion.*

We would love to perform this experiment, and have tried hard to achieve a germline specific rescue. Unfortunately, *utx-1* is an essential gene. Therefore, expressing a rescuing *utx-1* construct only in the germline will lead to severe developmental defects of the animals due to the lack of *utx-1* in other tissues (Vandamme et al. 2012, Figure 1). Nonetheless, and as stated in our previous response letter, we did try to express *utx-1* from *pie-1* or *gld-1* promoter but the animals turned sterile. Therefore, we resorted to the use of the *rrf-1* background. In *rrf-1* animals, neither the somatic gonad nor the DTC are able to stage an RNAi response. This is important with regard to the concern below.

While this approach has certain limitations – for example, *rrf-1* worms are able to stage an RNAi response in the intestine – it is the best we can do to investigate the autonomous versus non-autonomous roles of *utx-1* in GeCo (this new data is now shown in Figure 4—figure supplement 1).

*The conclusion that UTX-1 acts in the germ line is also not convincing for three reasons.*

*First, Seelk et al. use the promoter of utx-1 with a histone GFP readout (Figure 5). While this shows changes in glp and mes mutants, it is convincing with regard to UTX activity because the germ line relies heavily on post-transcriptional regulation. To ensure that the protein reflects RNA levels, the authors need to look at the endogenous protein or a single copy, rescuing reporter that has the whole protein and regulatory sequences.*

We would like to point out that this experiment has been already done (Figure 5—figure supplement 1). We expressed a single copy-integrated, FLAG and GFP-tagged UTX-1 protein from the endogenous promoter and under the control of the endogenous *utx-1* 3'UTR. This fusion protein rescued the *utx-1* mutant and was expressed in the germline in the same pattern as the *utx-1* promoter reporter. We realized that the labeling of this figure could have been misleading and changed it (from *putx-1::FLAG-GFP::utx-1*) to *"putx-1::flag-gfp-tev::utx-1::utx-1 3’utr".*

*Second, Vandamme 2012 argued that UTX-1 activity in the soma influenced the germ line, perhaps through the distal tip cell. The distal tip cell signals through the Notch pathway, so this is a reasonable idea.*

As discussed above, Kumsta and Hansen, 2012 showed that RNAi is defective in the somatic gonad, including DTC, in *rrf-1* mutants. We now show in Figure 4—figure supplement 1 that *utx-1* RNAi in the *rrf-1* background still inhibited GeCo , arguing against a role for UTX-1 in DTC or other cells of the somatic gonad.

*Third, Vandamme 2012 did not show endogenous UTX-1 expression in the germ line, as stated by Seelk et al.*

We apologize for this obvious mistake. Indeed, Vandamme et al., 2012 showed with antibody staining that UTX-1 is present in the germline precursor cell at the 4-cell stage. For other tissues, they used extrachromosomal arrays, which, as a rule, are silenced in the germline. With the citation removed, the paragraph “GLP-1_Notch_ and PRC2 regulate

expression of the H3K27 demethylase UTX-1” now reads:

“To test the potential expression of *utx-1* in the germ line, we constructed a strain expressing single copy-integrated, FLAG and GFP-tagged, functional UTX-1 (expressed from the endogenous *utx-1* promoter under the control of endogenous *utx-1* 3'UTR). We found that this protein was indeed expressed in the germline (Figure 5—figure supplement 1).”

*The key question is whether UTX and H3K27me3 function in the germ line and downstream of Notch to control reprogramming via target gene regulation. It may be that some events occur in the soma, so I don't think the authors need to test everything for cell autonomy, but utx-1 would be useful, particularly given previous publications suggesting at distal tip cell focus.*

As discussed above, Kumsta and Hansen, 2012 showed that RNAi is defective in the DTC in *rrf-1* mutants. We now show in Figure 4—figure supplement 1 that *utx-1* RNAi in the *rrf-1* background inhibited GeCo enhancement, arguing against a role for UTX-1 in DTC (or other cells of the somatic gonad).

*The H3K27me3 stains appear to have been done without controls, so it is impossible to judge if there are any changes in different strain backgrounds. (If there were, it would be more convincing that utx was acting in the germ line.) A well-controlled experiment would include an on-slide control (e.g. a marked N2) and a germline control (e.g. histone H3). The ratio of H3K27me3:H3 would give an indication of whether levels are changing.*

We would like to point out that we did include a wild-type control (Figure 2—figure supplement 4). The stainings were performed independently by both Ciosk and Tursun labs, using whole worms or dissected gonads. The wild-type samples were not stained on the same slide as mutants, but were processed in the same batch and staining was always consistent. The genomic data suggested that increased GLP-1_Notch_ signaling actives a subset of PRC2-repressed genes, so the purpose of this experiment was to confirm that there was no global loss of H3K27me3 upon increased GLP-1_Notch_ signaling. Indeed, we found that, in the tested mutants, the levels and overall appearance of H3K27me3 were not obviously altered compared to controls.

*References to be added:*

*Introduction, second paragraph: add Horner et al., 1998 as one of the first examples of reprogramming in an intact organism and of timing of competence.*

Introduction, second or third paragraph: add Yuzyuk 2009 for PRC2 -dependent termination of plasticity and promotion of cell fate stabilization.

*Introduction, third paragraph: add Maures 2011, Jin 2011, maybe Vandamme 2012 for identification of utx-1 as a regulator of H3K27me3.*

We added these citations at the suggested positions.

*Reviewer #2:*

*A number of changes need to be made to the manuscript so that the results and possible conclusion are more transparent.*

1) Paragraph with heading “Both GLP-1^Notch^ and PRC2 regulate expression of the H3K27 demethylase UTX-1”:

*A) The authors should first explain what is the expected expression pattern for a GLP-1 transcriptional target, based on the known and experimentally validated direct transcriptional targets of GLP-1 Notch in the gonad, lst-1 and sygl-1, described in Kershner et al. 2015. The failure to do so makes the paper hard to understand for readers outside the immediate C. elegans field.*

We re-wrote this paragraph and, before describing the *utx-1* expression pattern, introduce the patterns of *sygl-1* and *lst-1*, as suggested:

“With both the UTX-1 fusion protein and the putx-1 reporter, we expected expression patterns similar to that of other reported GLP-1_Notch_ target genes (Lamont et al. 2004; Lee et al. 2006; Kershner et al. 2014). […] By contrast, both the UTX-1 fusion protein and the *putx-1* reporter were little or not expressed in the distal-most, proliferative part of the germ line (Figure 5; Figure 5—figure supplement 1).”

*B) The expected pattern based on the work of Kershner et al., 2015, is that GLP-1 dependent transcripts are limited to the distal most 5 to 10 cell diameters, consistent with ligand expression limited to the distal tip cell. The authors interpretation of the putx-1::gfp (Figure 1) and putx-1::FLAG-GFP::utx-1 (Figure 5—figure supplement 1) is that there is weak expression in the distal-most region of the germline where GLP-1 targets are expected to be expressed. However, this reviewer cannot see any putx-1 GFP signal above background, and this view is supported by the absence of mRNA in the distal-most germline from the in situ hybridization experiments in Figure 5—figure supplement 2). Therefore, the sentence "Consistent with the expression patterns of the endogenous utx-1 (Vandamme et al. 2012) and of a GFP used, functional utx-1 transgene (Figure 5—figure supplement 1), the utx-1 reporter was weakly expressed in the distal-most, proliferative part of the germline (Figure 5)." should be changed to indicate that there is "little or no expression in the distal-most, proliferative part of the germline".*

This sentence has been modified accordingly:

“By contrast, both the UTX-1 fusion protein and the *putx-1* reporter were little or not expressed in the distal-most, proliferative part of the germ line (Figure 5; Figure 5—figure supplement 1).”

*C) The sentence "However, a similar expression pattern has been reported for another reported GLP-1Notch target gene, lip-1 (Hajnal and Berset 2002; Lee et al. 2006)." should be removed. Hajnal and Berset 2002 showed that lip-1 is a lin-12 transcriptional target in vulval development, with no information on glp-1 or the germline. Lee et al. 2006 did not perform the glp-1 dependence experiment comparing gld-1 gld-2 with gld-1 gld-2; glp-1, which the authors in this manuscript performed, and thus it is uncertain if lip-1 is a germline target of GLP-1 in distal-most germ cells.*

This sentence has been removed, as requested. We mention now the expression pattern of *lip-1* in the Discussion, citing Lee et al., 2006.

*D) In the first two paragraphs of the subsection “UTX-1 is required for GLP-1^Notch^-mediated GeCo enhancement”. There is no evidence that GLP-1 signaling occurs in the medial/proximal gonad, as GLP-1 protein is not found in this region (Crittenden et al., 1994). Therefore, the expression of the 33 genes indicated is either an indirect consequence of GLP-1 signaling (another transcription factor that is transcribed in the distal-most germline stem cells in response to GLP-1 signaling which then acts to induce transcription much later in development, in pachytene cells) or is not regulated by GLP-1 signaling at all. The sentence needs to be changed to indicate that the observed expression is not consistent with direct GLP-1 transcriptional regulation.*

The sentence has been changed as recommended:

"When examining the existing mRNA hybridization patterns of GLP-1_Notch_-activated and PRC2-repressed genes (33 genes), we noticed that half of these (18, all X-linked) are similarly expressed in the medial and/or proximal, but not the distal-most, gonads ([Supplementary-material SD6-data]), arguing against direct transcriptional activation of these genes by GLP-1_Notch_ in the wild type."

*2) In the subsection “Both GLP-1^Notch^ and PRC2 regulate expression of the H3K27 demethylase UTX-1”. "Deleting those sites drastically reduced reporter expression in the medial and proximal portions of the gonad (Figure 5)." Figure 5 indicates that the decrease is at most 25%. The sentence should be "Deleting those sites reduced reporter expression by approximately 25% in the medial and proximal portions of the gonad (Figure 5)."*

The sentence has been changed as recommended:

"[…]the activation of the *utx-1* promoter by Notch signaling was depended on the putative LAG-1/CSL binding sites within the promoter (Yoo et al. 2004), as deleting those sites reduced reporter expression by approximately one-fourth (Figure 5)."

*3) The last paragraph of the subsection “Both GLP-1^Notch^ and PRC2 regulate expression of the H3K27 demethylase UTX-1”.*

*"Overall, our observations suggest that PRC2 and GLP-1^Notch^ signaling have antagonistic effects on utx-1 transcription. However, the endogenous levels of GLP-1^Notch^ signaling are apparently insufficient to overcome PRC2-mediated repression in the distal part of the gonad" does not emphasize the findings in the manuscript and should be rewritten in the following way:*

*"Overall, our observations suggest that PRC2 and GLP-1^Notch^ signaling have antagonistic effects on utx-1 transcription that can be observed in a PRC2 mutant background. However, in wild type the endogenous levels of GLP-1^Notch^ signaling are apparently insufficient to overcome PRC2-mediated repression in the distal part of the gonad".*

This paragraph has been rephrased as suggested:

“Summarizing, based on the observations in mutant backgrounds, PRC2 and GLP-1_Notch_ signaling have antagonistic effects on *utx-1* transcription. However, in wild type, the endogenous levels of GLP-1_Notch_ signaling are apparently insufficient to overcome PRC2-mediated repression of *utx-1* in the distal-most part of the gonad.”

*4) Discussion. While the sentence "This suggests that the identified set of genes may contain additional physiological GLP-1Notch targets." may be true, of the genes analyzed in this manuscript, none have been shown to be convincing GLP-1^Notch^ targets in wild type animals.*

*The Discussion should instead focus on their novel findings – that PRC2 is apparently acting antagonistically to prevent certain GLP-1^Notch^ targets from being expressed in the germline, which makes sense as it is likely important to block the expression of some somatic GLP-1 targets from being expressed in germ cells, and that these include utx-1 which promotes reprogramming.*

We have re-written the Discussion, according to these suggestions, to our best ability.

*Reviewer #3:*

*I think in general the study stands up as providing evidence that there is some antagonism between Notch activation and PRC2-mediated repression of genes, and that enhancement of Notch activity in germ cells yields similar transcriptional consequences to loss of PRC2 repression in germ cells – enhanced X-linked gene expression. Similarly, these conditions also both make it easier to drive germ cells towards somatic differentiation upon ectopic activation of somatic fate drivers, such as che-1.*

*The question still stands whether this is informative for either normal germ cell development or for understanding the balance of activation and repression controlling early embryo development or programming/reprogramming.*

We are happy to hear that our Reviewer likes the evidence we provide for the antagonistic relationship of Notch and PRC2 in the germline.

As also requested by Reviewer 2, the Discussion now hopefully states clearly that, at this point, the connection to the wild-type development remains an open question.

*I think it's clear that the processes studied are artificial in terms of germ cell development – but not entirely so. I think the authors do show that the balance of Glp-1 signaling and PRC2 repression are essential for normal germ cell development. I also think they've shown they can uncouple some of the Glp-1-dependent effects from proliferation phenotypes. This probably underlies the differences in absolute phenotypes between Glp-1 gof and PRC2 mutants (which are presumably maternally rescued mutants): proliferation defects in the former compared to little visible defects in the maternally rescued germ cells. It's formally interesting that raising the level of Notch activity has the same effect on transcriptional regulation as lowering PRC2 activity in these cells. It's interesting that this seems to be separate from any need for proliferation to be dysregulated.*

*I think the authors have adequately addressed the technical concerns about this conclusion.*

*There still is a bit of what I view as semantic issues with how some things are phrased that can be over-interpreted by the reader, or may indeed still be overstatements by the authors:*

*For example, the subsection “GLP-1^Notch^ and PRC2 have antagonistic functions in the germline” states "that, in the C. elegans gonad, specific PRC2-repressed genes are activated via GLP-1^Notch^ signaling." This certainly implies that glp-1 activation of genes like utx-1 is part of normal germ cell function, which is unlikely to be true.*

We have rephrased this sentence to: "… suggests that increased GLP-1 signaling can induce expression of specific PRC2-repressed genes."

*Lastly, the description of the mes mutants used needs to be clarified. I am assuming that when the authors are referring to mes-2(bn11) mutants, etc. they are actually referring to the F1 maternally rescued homozygote offspring (e.g., M Z ). It is important to clarify that these animals and their germ cells are not completely devoid of a history of PRC2 activity, which is why they have germ cells to count and look at in the first place. Absent maternal MES protein, there wouldn't be but a few dying germ cells in the adult gonads to look at.*

We thank the Reviewer for pointing this out. Indeed, we used M Z- animals. We now indicate it in the main text:

"At 20°C, both the temperature-sensitive gain-of-function *glp-1(ar202)* and the homozygous loss-of-function *mes-2(bn11)* mutants, derived from heterozygous mothers providing maternal MES-2 (*mes-2* M Z- mutants), were viable and produced gonads with nearly wild-type appearance. The double *mes-2* M Z-; *glp-1(ar202)* mutants, however, displayed distal and proximal tumors at the same temperature[…]"

"[…]we first determined putative PRC2 targets by expression analyses on isolated wild- type, M Z- *mes-2* or *mes-6* mutant gonads…"

[…]in the legend to Figure 2:

"The *mes-2(bn11)* M Z- single mutant gonads have wild-type appearance at 20°C."

"At the same temperature, *mes-2(bn11)* M Z-; *glp-1(ar202)* double mutants developed extensive germ line tumors in 32/32 of the examined gonads."

"The plots correlate changes in gene expression in M Z- *mes-2* mutants with changes in gene expression changes in M Z- *mes-6* mutants."

[…]and in the in the legend to Figure 5—figure supplement 2:

"In M Z- *mes-6(bn66)* mutants, *utx-1* is upregulated throughout the gonads compared to the control wild-type gonads."